# Anti-Backdoor Learning: Training Clean Models on Poisoned Data

**Yige Li**
Xidian University
yglee@stu.xidian.edu.cn

**Xixiang Lyu** [†]
Xidian University
xxlv@mail.xidian.edu.cn

**Nodens Koren**
University of Copenhagen
nodens.f.koren@di.ku.dk

**Lingjuan Lyu**
Sony AI
Lingjuan.Lv@sony.com

**Bo Li**
University of Illinois at Urbana–Champaign
lbo@illinois.edu

**Xingjun Ma** [†]
Fudan University
danxjma@gmail

## Abstract

Backdoor attack has emerged as a major security threat to deep neural networks (DNNs). While existing defense methods have demonstrated promising results on detecting or erasing backdoors, it is still not clear whether robust training methods can be devised to prevent the backdoor triggers being injected into the trained model in the first place. In this paper, we introduce the concept of *anti-backdoor learning*, aiming to train *clean* models given backdoor-poisoned data. We frame the overall learning process as a dual-task of learning the *clean* and the *backdoor* portions of data. From this view, we identify two inherent characteristics of backdoor attacks as their weaknesses: 1) the models learn backdoored data much faster than learning with clean data, and the stronger the attack the faster the model converges on backdoored data; 2) the backdoor task is tied to a specific class (the backdoor target class). Based on these two weaknesses, we propose a general learning scheme, Anti-Backdoor Learning (ABL), to automatically prevent backdoor attacks during training. ABL introduces a two-stage *gradient ascent* mechanism for standard training to 1) help isolate backdoor examples at an early training stage, and 2) break the correlation between backdoor examples and the target class at a later training stage. Through extensive experiments on multiple benchmark datasets against 10 state-of-the-art attacks, we empirically show that ABL-trained models on backdoor-poisoned data achieve the same performance as they were trained on purely clean data. Code is available at https://github.com/bboylyg/ABL.

## 1 Introduction

A backdoor attack is a type of training-time data poisoning attack that implant backdoor triggers into machine learning models by injecting the trigger pattern(s) into a small proportion of the training data [1]. It aims to trick the model to learn a strong but task-irrelevant correlation between the trigger pattern and a target class, and optimizes three objectives: stealthiness of the trigger pattern, injection (poisoning) rate and attack success rate. A backdoored model performs normally on clean test data yet consistently predicts the target class whenever the trigger pattern is attached to a test example.

---

[†]Correspondence to: Xixiang Lyu, Xingjun Ma.

35th Conference on Neural Information Processing Systems (NeurIPS 2021).

Studies have shown that deep neural networks (DNNs) are particularly vulnerable to backdoor attacks [2]. Backdoor triggers are generally easy to implant but hard to detect or erase, posing significant security threats to deep learning.

Existing defense methods against backdoor attacks can be categorized into two types: detection methods and erasing methods [3]. Detection methods exploit activation statistics or model properties to determine whether a model is backdoored [4, 5], or whether a training/test example is a backdoor example [6, 7]. While detection can help identify potential risks, the backdoored model still needs to be purified. Erasing methods [8–10] take one step further and remove triggers from the backdoored model. Despite their promising results, it is still unclear in the current literature whether the underlying model learns clean and backdoor examples in the same way. The exploration of this aspect leads to a fundamental yet so far overlooked question, "*Is it possible to train a clean model on poisoned data?*"

Intuitively, if backdoored data can be identified during training, measures can be taken to prevent the model from learning them. However, we find that this is not a trivial task. One reason is that we do not know the proportion nor the distribution of the backdoored data in advance. As shown in Figure 1, on CIFAR-10, even if the poisoning rate is less than 1%, various attacks can still achieve high attack success rates. This significantly increases the difficulty of backdoor data detection as the model's learning behavior may remain the same with or without a few training examples. Even worse, we may accidentally remove a lot of valuable data when the dataset is completely clean. Another important reason is that the backdoor may have already been learned by the model even if the backdoor examples are identified at a later training stage.

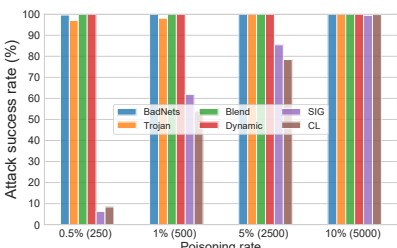

Figure 1: Attack success rate (ASR) of 6 backdoor attacks under different poisoning rates on CIFAR-10. 4 out of the 6 attacks can achieve nearly 100% ASR at poisoning rate 0.5%.

In this paper, we frame the overall learning process on a backdoor-poisoned dataset as a dual-task learning problem, with the learning of the clean portion as the original (clean) task and the learning of the backdoored portion as the backdoor task. By investigating the distinctive learning behaviors of the model on the two tasks, we identify two inherent characteristics of backdoor attacks as their weaknesses. **First**, the backdoor task is a much easier task compared to the original task. Consequently, the training loss of the backdoored portion drops abruptly in early epochs of training, whereas the loss of clean examples decreases at a steady pace. We also find that the stronger the attack, the faster the loss on backdoored data drops. This finding indicates that the backdoor correlations imposed by stronger attacks are easier and faster to learn, and marks one distinctive learning behavior on backdoored data. **Second**, the backdoor task is tied to a specific class (i.e., the backdoor target class). This indicates that the correlation between the trigger pattern and the target class could be easily broken by simply randomizing the class target, for instance, shuffling the labels of a small proportion of examples with low loss.

Inspired by the above observations, we propose a principled *Anti-Backdoor Learning (ABL)* scheme that enables the training of clean models without any prior knowledge of the distribution of backdoored data in datasets. ABL introduces a *gradient ascent* based anti-backdoor mechanism into the standard training to help isolate low-loss backdoor examples in early training and unlearn the backdoor correlation once backdoor examples are isolated. In summary, our main contributions are:

- We present a novel view of the problem of robust learning with poisoned data and reveal two inherent weaknesses of backdoor attacks: faster learning on backdoored data and target-class dependency. The stronger the attack is, the more easily it can be detected or disrupted.

- We propose a novel Anti-Backdoor Learning (ABL) method that is capable of training clean models on poisoned data. To the best of our knowledge, ABL is the *first* method of its kind in the backdoor defense literature, complementing existing defense methods.

- We empirically show that our ABL is robust to 10 state-of-the-art backdoor attacks. The models trained using ABL are of almost the same clean accuracy as they were directly trained on clean data and the backdoor attack success rates on these models are close to random guess.

## 2 Related Work

**Backdoor Attack.** Existing backdoor attacks aim to optimize three objectives: 1) making the trigger pattern stealthier; 2) reducing the poisoning (injection) rate; 3) increasing the attack success rate [3]. Creative design of trigger patterns can help with the stealthiness of the attack. These can be simple patterns such as a single pixel [6] and a black-white checkerboard [1], or more complex patterns such as blending backgrounds [11], natural reflections [12], invisible noise [13–16], adversarial patterns [17] and sample-wise patterns [18, 19]. Backdoor attacks can be further divided into two categories: dirty-label attacks [1, 11, 12] and clean-label attacks [20–22, 17, 16]. Clean-label attacks are arguably stealthier as they do not change the labels. Backdoor attackers can also inject backdoors via retraining the victim model on a reverse-engineered dataset without accessing the original training data [23]. Most of these attacks can achieve a high success rate (e.g., $> 95\%$) by poisoning only 10% or even less of the training data. A recent study by Zhao *et al.*[24] showed that even models trained on clean data can have backdoors, highlighting the importance of anti-backdoor learning.

**Backdoor Defense.** Existing backdoor defenses fall under the categories of either detection or erasing methods. Detection methods aim to detect anomalies in input data [7, 6, 25–28] or whether a model is backdoored [4, 5, 29, 30]. These methods typically show promising accuracies; however, the potential impact of backdoor triggers remains uncleared in the backdoored models. On the other hand, erasing methods take a step further and aim to purify the adverse impacts on models caused by the backdoor triggers. The current state-of-the-art erasing methods are Mode Connectivity Repair (MCR) [9], Neural Attention Distillation (NAD) [10] and Adversarial Neuron Pruning (ANP) [31]. MCR mitigates the backdoors by selecting a robust model in the path of loss landscape, NAD leverages attention distillation to erase triggers, while ANP prunes adversarially sensitive neurons to purify the model. Other previous methods, including finetuning, denoising, and fine-pruning [8], have been shown to be insufficient against the latest attacks [32, 33, 12]. An early work [34] found that DNNs are more accurate on clean samples in an early training stage, while we find that the backdoor attacks studied in [34] are only limited to simple BadNets attacks. Thus, it would be interesting to further study the different properties of diverse types of backdoor attacks.

In this paper, we introduce the concept of *anti-backdoor learning*. Unlike existing methods, our goal is to train clean models directly on poisoned datasets without further altering the models or the input data. This requires a more in-depth understanding of the distinctive learning behaviors on backdoored data. However, such information is not available in the current literature. Anti-backdoor learning methods may replace the standard training to prevent potential backdoor attacks in real-world scenarios where data sources are not 100% reliable, and the distribution or even the presence of backdoor examples are unknown.

## 3 Anti-Backdoor Learning

In this section, we first formulate the Anti-Backdoor Learning (ABL) problem, then reveal the distinctive learning behaviors on clean versus backdoor examples and introduce our proposed ABL method. Here, we focus on classification tasks with deep neural networks.

**Defense Setting.** We assume the backdoor adversary has pre-generated a set of backdoor examples and has successfully injected these examples into the training dataset. We also assume the defender has full control over the training process but has no prior knowledge of the proportion nor distribution of the backdoor examples in a given dataset. The defender's goal is to train a model on the given dataset (potentially poisoned) that is as good as models trained on purely clean data. Moreover, if an isolation method is used, the defender may identify only a subset of the backdoor examples. For instance, in the case of 10% poisoning, the isolation rate might only be 1%. Robust learning methods developed under our defense setting could benefit companies, research institutes, government agencies or MLaaS (Machine Learning as a Service) providers to train backdoor-free models on potentially poisoned data. More explanations about our threat model and how our proposed ABL method can help other defense settings can be found in Appendix B.10.

**Problem Formulation.** Consider a standard classification task with a dataset $\mathcal{D} = \mathcal{D}_c \cup \mathcal{D}_b$ where $\mathcal{D}_c$ denotes the subset of clean data and $\mathcal{D}_b$ denotes the subset of backdoor data. The standard training

trains a DNN model $f_\theta$ by minimizing the following empirical error:

$$\mathcal{L} = \mathbb{E}_{(\boldsymbol{x},y)\sim\mathcal{D}}[\ell(f_\theta(\boldsymbol{x}),y)] = \underbrace{\mathbb{E}_{(\boldsymbol{x},y)\sim\mathcal{D}_c}[\ell(f_\theta(\boldsymbol{x}),y)]}_{\text{clean task}} + \underbrace{\mathbb{E}_{(\boldsymbol{x},y)\sim\mathcal{D}_b}[\ell(f_\theta(\boldsymbol{x}),y)]}_{\text{backdoor task}}, \qquad (1)$$

where $\ell(\cdot)$ denotes the loss function such as the commonly used cross-entropy loss. The overall learning task is decomposed into two tasks where the first *clean task* is defined on the clean data $\mathcal{D}_c$ while the second *backdoor task* is defined on the backdoor data $\mathcal{D}_b$. Since backdoor examples are often associated with a particular target class, all data from $\mathcal{D}_b$ may share the same class label. The above decomposition indicates that the standard learning approach tends to learn both tasks, resulting in a backdoored model.

To prevent backdoor examples from being learned, we propose anti-backdoor learning to minimize the following empirical error instead:

$$\mathcal{L} = \mathbb{E}_{(\boldsymbol{x},y)\sim\mathcal{D}_c}[\ell(f_\theta(\boldsymbol{x}),y)] - \mathbb{E}_{(\boldsymbol{x},y)\sim\mathcal{D}_b}[\ell(f_\theta(\boldsymbol{x}),y)]. \qquad (2)$$

Note the maximization of the backdoor task is defined on $\mathcal{D}_b$. Unfortunately, the above objective is undefined during training since we do not know the $\mathcal{D}_b$ subset. Intuitively, $\mathcal{D}_b$ can be detected and isolated during training if the model exhibits an atypical learning behavior on the backdoor examples. In the following subsection, we will introduce one such behavior, which we recognize as the first weakness of backdoor attacks.

### 3.1 Distinctive Learning Behaviors on Backdoor Examples

We apply 6 classic backdoor attacks including BadNets [1], Trojan [23], Blend [11], Dynamic [18], SIG [35], and CL [21], and 3 feature-space attacks including FC [20], DFST [36], and LBA [32] to poison 10% of CIFAR-10 training data. We train a WideResNet-16-1 model [37] on the corresponding poisoned dataset using the standard training method by solving equation (1) for each attack. Each model is trained following the standard settings (see Section 4 and Appendix A.2). We plot the average training loss (i.e., cross-entropy) on clean versus backdoored training examples in Figure 2. Clearly, for all 9 attacks, the training loss on backdoor examples drops much faster than that on clean examples in the first few epochs. Both pixel- and feature-space attacks exhibit this faster-learning pattern consistently, although some feature-space attacks (FC and LBA) can slow down the process to some extent. For all attacks except SIG, FC and LBA, the training loss reaches almost zero after only two epochs of training. Moreover, according to the attack success rate, the stronger the attack is, the faster the training loss on backdoor examples drops. More results on GTSRB and an ImageNet subset can be found in Appendix B.2.

The above observation indicates that the backdoor task is much easier than the clean task. This is not too surprising. In a typical clean dataset, not all examples are easy examples. Thus, it requires a certain number of training epochs to minimize the loss on those examples, even for small datasets like CIFAR-10. On the contrary, a backdoor attack adds an explicit correlation between the trigger pattern and the target class to simplify and accelerate the injection of the backdoor trigger. We argue that this is a fundamental requirement and also a major weakness of backdoor attacks. For a backdoor attack to work successfully, the trigger(s) should be easily learnable by the model, or else the attack would lose its effectiveness or require a much higher injection rate, which goes against its key objectives. Therefore, the stronger the attack is, the faster the training loss on backdoor examples drops to zero; e.g., compare FC with other attacks in Figure 2. We also show in Figure 6 in Appendix B.1 that the training loss of the backdoor task drops more rapidly as we increase the poisoning rate.

Based on the above observation, one may wonder if backdoor examples can be easily removed by filtering out the low-loss examples at an early stage (e.g., the 5th epoch). However, we find that this strategy is ineffective for two reasons. First, the training loss in Figure 2 is the average training loss which means some backdoor examples can still have high training loss. Additionally, several powerful attacks such as Trojan and Dynamic can still succeed even with very few (50 or 100) backdoor examples. Second, if the training progresses long enough (e.g., beyond epoch 20), many clean examples will also have a low training loss, which makes the filtering significantly inaccurate. Therefore, we need a strategy to amplify the difference in training loss between clean and backdoor examples. Moreover, we need to unlearn the backdoor since the backdoor examples can only be identified when they are learned into the model (i.e., low training loss).

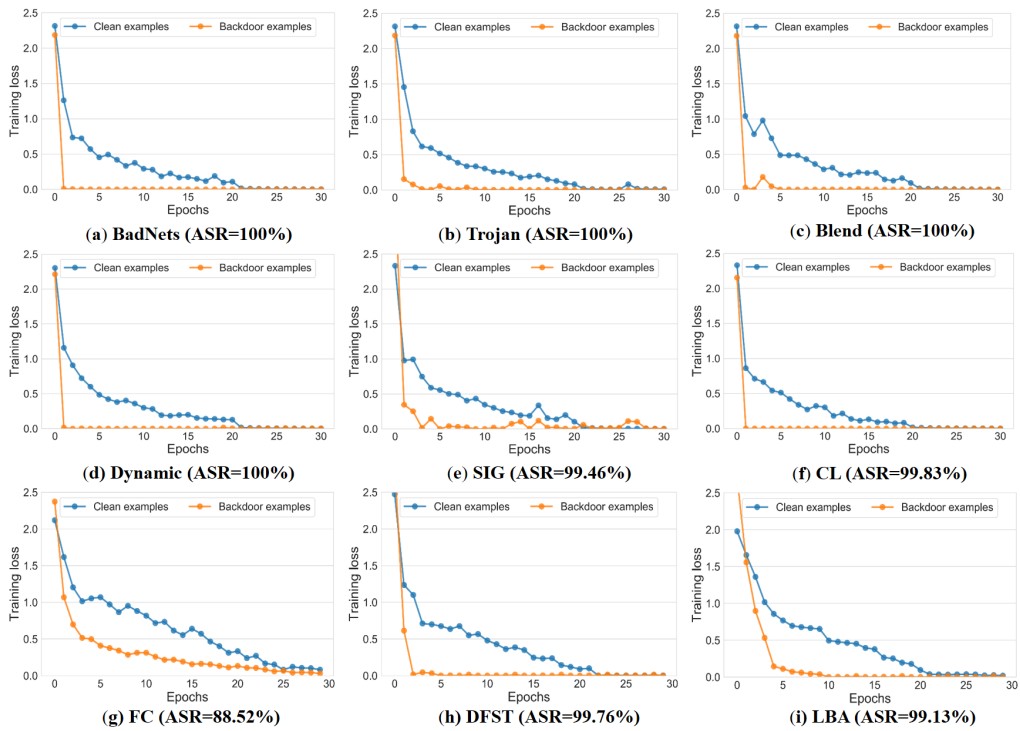

Figure 2: The training loss on clean versus backdoor examples crafted by 9 backdoor attacks including BadNets [1], Trojan [23], Blend [11], Dynamic [18], SIG [35], and CL [21], FC [20], DFST [36], and LBA [32]. This experiment is conducted with WideResNet-16-1 [37] on CIFAR-10 under poisoning rate 10%. ASR: attack success rate (on WideResNet-16-1).

## 3.2 Proposed Anti-Backdoor Learning Method

Suppose the total number of training epochs is $T$, we decompose the entire training process into two stages, i.e., early training and later training. We denote the turning epoch from early training to later training by $T_{te}$. Our anti-backdoor learning method consists of two key techniques: 1) *backdoor isolation* during early training, and 2) *backdoor unlearning* during later training. The turning epoch is chosen to be the epoch where the average training loss stabilizes at a certain level.

**Backdoor Isolation.** During early training, we propose a *local gradient ascent* (LGA) technique to trap the loss value of each example around a certain threshold $\gamma$. We use the loss function $\mathcal{L}_{\text{LGA}}$ in equation (3) to achieve this. The gradient ascent is said to be "local" because the maximization is performed around a fixed loss value $\gamma$. In other words, if the loss of a training example goes below $\gamma$, gradient ascent will be activated to boost its loss to $\gamma$; otherwise, the loss stays the same. Doing so will force backdoor examples to escape the $\gamma$ constraint since their loss values drop significantly faster. The choice of an appropriate $\gamma$ lies in the core of this strategy, as an overly large $\gamma$ will hurt the learning of the clean task, while an overly small $\gamma$ may not be strong enough to segregate the clean task from the backdoor task. Note that $\gamma$ can be determined by the strength of potential attacks: stronger attacks only need a smaller $\gamma$ to isolate. Since most backdoor attacks are strong, the poisoned data can easily reach a small loss value below 0.5. So, we set $\gamma = 0.5$ in our experiments and show its consistent performance across different datasets and models in Section 4.2 and Appendix B.5. At the end of early training, we segregate examples into disjoint subsets: $p$ percent of data with the lowest loss values will be isolated into the backdoor set $\widehat{\mathcal{D}}_b$ ($p = |\widehat{\mathcal{D}}_b|/|\mathcal{D}|$), and the rest into the clean set $\widehat{\mathcal{D}}_c$ ($\mathcal{D} = \widehat{\mathcal{D}}_b \cup \widehat{\mathcal{D}}_c$). An important note here is that the isolation rate (e.g., $p = 1\%$) is assumed to be much smaller than the poisoning rate (e.g., 10%).

**Backdoor Unlearning.** With the clean and backdoor sets, we can then proceed with the later training. Note that at this stage, the backdoor has already been learned by the model. Given the above low isolation rate, an effective backdoor unlearning method is required to make the model unlearn the backdoor with a small subset $\widehat{\mathcal{D}}_b$ of backdoor examples while simultaneously learning the remaining

(unisolated) backdoor examples in the clean set $\widehat{\mathcal{D}}_c$. We make this possible by exploiting the second weakness of backdoor attacks: the backdoor trigger is usually associated with a particular backdoor target class. We propose to use the loss $\mathcal{L}_{\text{GGA}}$ defined in equation (2) for this purpose. In $\mathcal{L}_{\text{GGA}}$, a *global gradient ascent* (GGA) is defined on the isolated subset $\widehat{\mathcal{D}}_b$. Unlike the local gradient ascent, it is not constrained to be around a fixed loss value. We will show in Section 4.2 that a low isolation rate of 1% is able to effectively unlearn the backdoor against 50% poisoning.

The loss functions used by our ABL for its two training stages are summarized as follows,

$$
\mathcal{L}_{\text{ABL}}^t = \begin{cases} \mathcal{L}_{\text{LGA}} = \mathbb{E}_{(\boldsymbol{x},y)\sim\mathcal{D}}\big[\,\text{sign}(\ell(f_\theta(\boldsymbol{x}),y) - \gamma) \cdot \ell(f_\theta(\boldsymbol{x}),y)\big] & \text{if } 0 \le t < T_{te} \\ \mathcal{L}_{\text{GGA}} = \mathbb{E}_{(\boldsymbol{x},y)\sim\widehat{\mathcal{D}}_c}\big[\ell(f_\theta(\boldsymbol{x}),y)\big] - \mathbb{E}_{(\boldsymbol{x},y)\sim\widehat{\mathcal{D}}_b}\big[\ell(f_\theta(\boldsymbol{x}),y)\big] & \text{if } T_{te} \le t < T, \end{cases}
\tag{3}
$$

where $t \in [0, T-1]$ is the current training epoch, $\text{sign}(\cdot)$ is the sign function, $\gamma$ is the loss threshold for LGA and $\widehat{\mathcal{D}}_b$ is the isolated backdoor set with the isolation rate $p = |\widehat{\mathcal{D}}_b|/|\mathcal{D}|$. During early training ($0 \le t < T_{te}$), the loss will be automatically switched to $-\ell(f_\theta(\boldsymbol{x}),y)$ if $\ell(\cdot)$ is smaller than $\gamma$ by the sign function; otherwise the loss stays the same, i.e., $\ell(f_\theta(\boldsymbol{x}),y)$. Note that $\mathcal{L}_{\text{LGA}}$ loss may also be achieved by the flooding loss proposed in [38] to prevent overfitting: $|\ell(f_\theta(\boldsymbol{x}),y) - b| + b$ where $b$ is a flooding parameter. We would like to point out that LGA serves only one part of our ABL and can potentially be replaced by other backdoor detection methods. Additionally, we will show that a set of other techniques may also achieve backdoor isolation and unlearning, but they are far less effective than our ABL (see Section 4.3 and Appendix B.3).

# 4    Experiments

**Attack Configurations.** We consider 10 backdoor attacks in our experiments, including four dirty-label attacks: BadNets [1], Trojan attack [23], Blend attack [11], Dynamic attack [18], two clean-label attacks: Sinusoidal signal attack(SIG) [35] and Clean-label attack(CL) [21], and four feature-space attacks: Feature collision (FC) [20], Deep Feature Space Trojan Attack (DFST) [24], Latent Backdoor Attack (LBA) [32], and Composite Backdoor Attack (CBA) [39]. We follow the settings suggested by [10] and the open-sourced code corresponding to their original papers to configure these attack algorithms. All attacks are evaluated on three benchmark datasets, CIFAR-10 [40], GTSRB [41] and an ImageNet subset [42], with two classical model structures including WideResNet (WRN-16-1) [37] and ResNet-34 [43]. No data augmentations are used for these attacks since they hinder the backdoor effect [12]. To keep their original configurations of dataset and parameter settings, here we only run the four feature-space attacks on CIFAR-10 dataset. We also omit some attacks on GTSRB and ImageNet datasets due to a failure of reproduction following their original papers. The detailed settings of the 10 backdoor attacks are summarized in Table 5 (see Appendix A.2).

**Defense and Training Details.** We compare our ABL with three state-of-the-art defense methods: Fine-pruning (FP) [8], Mode Connectivity Repair (MCR) [9], and Neural Attention Distillation (NAD) [10]. For FP, MCR and NAD, we follow the configurations specified in their original papers, including the available clean data for finetuning/repair/distillation and training settings. The comparison with other data isolation methods are shown in Section 4.3. For our ABL, we set $T = 100$, $T_{te} = 20$, $\gamma = 0.5$ and an isolation rate $p = 0.01$ (1%) in all experiments. The exploration of different $T_{te}$, $\gamma$, and isolation rates $p$ are also provided in Section 4.1. Three data augmentation techniques suggested in [10] including random crop (padding = 4), horizontal flipping, and cutout, are applied for all defense methods. More details on defense settings can be found in Appendix A.3.

**Evaluation Metrics.** We adopt two commonly used performance metrics: Attack Success Rate (ASR), which is the classification accuracy on the backdoor test set, and Clean Accuracy (CA), the classification accuracy on clean test set.

## 4.1    Effectiveness of Our ABL Defense

**Comparison to Existing Defenses.** Table 1 demonstrates our proposed ABL method on CIFAR-10, GTSRB, and an ImageNet Subset. We consider 10 state-of-the-art backdoor attacks and compare the performance of ABL with three other backdoor defense techniques. It is clear that our ABL achieves the best results on reducing ASR against most of backdoor attacks, while maintaining an extremely high CA across all three datasets. In comparison to the best baseline method NAD, our ABL achieves 12.71% (7.69% vs. 20.40%), 11.90% (7.27% vs. 19.17%), and 7.35% (6.00% vs. 13.35%) lower

Table 1: The attack success rate (ASR %) and the clean accuracy (CA %) of 4 backdoor defense methods against 10 backdoor attacks including 6 classic backdoor attacks and 4 feature-space attacks. *None* means the training data is completely clean.

| Dataset | Types | No Defense | | FP | | MCR | | NAD | | ABL (Ours) | |
|---|---|---|---|---|---|---|---|---|---|---|---|
| | | ASR | CA | ASR | CA | ASR | CA | ASR | CA | ASR | CA |
| CIFAR-10 | *None* | 0% | 89.12% | 0% | 85.14% | 0% | 87.49% | 0% | 88.18% | 0% | **88.41%** |
| | BadNets | 100% | 85.43% | 99.98% | 82.14% | 3.32% | 78.49% | 3.56% | 82.18% | **3.04%** | **86.11%** |
| | Trojan | 100% | 82.14% | 66.93% | 80.17% | 23.88% | 76.47% | 18.16% | 80.23% | **3.81%** | **87.46%** |
| | Blend | 100% | 84.51% | 85.62% | 81.33% | 31.85% | 76.53% | **4.56%** | 82.04% | 16.23% | **84.06%** |
| | Dynamic | 100% | 83.88% | 87.18% | 80.37% | 26.86% | 70.36% | 22.50% | 74.95% | **18.46%** | **85.34%** |
| | SIG | 99.46% | 84.16% | 76.32% | 81.12% | 0.14% | 78.65% | 1.92% | 82.01% | **0.09%** | **88.27%** |
| | CL | 99.83% | 83.43% | 54.95% | 81.53% | 19.86% | 77.36% | 16.11% | 80.73% | **0%** | **89.03%** |
| | FC | 88.52% | 83.32% | 69.89% | 80.51% | 44.43% | 77.57% | 58.68% | 81.23% | **0.08%** | **82.36%** |
| | DFST | 99.76% | 82.50% | 78.11% | 80.23% | 39.22% | 75.34% | 35.21% | 78.40% | **5.33%** | **79.78%** |
| | LBA | 99.13% | 81.37% | 54.43% | 79.67% | 15.52% | 78.51% | 10.16% | 79.52% | **0.06%** | **80.52%** |
| | CBA | 90.63% | 84.72% | 77.33% | 79.15% | 38.76% | 76.36% | 33.11% | 82.40% | **29.81%** | **84.66%** |
| | Average | 97.73% | 83.55% | 75.07% | 80.62% | 24.38% | 76.56% | 20.40% | 80.37% | **7.69%** | **84.76%** |
| GTSRB | *None* | 0% | 97.87% | 0% | 90.14% | 0% | 95.49% | 0% | 95.18% | 0% | **96.41%** |
| | BadNets | 100% | 97.38% | 99.57% | 88.61% | 1.00% | 93.45% | 0.19% | 89.52% | **0.03%** | **96.01%** |
| | Trojan | 99.80% | 96.27% | 93.54% | 84.22% | 2.76% | 92.98% | 0.37% | 90.02% | **0.36%** | **94.95%** |
| | Blend | 100% | 95.97% | 99.50% | 86.67% | **6.83%** | 92.91% | 8.10% | 89.37% | 24.59% | **93.14%** |
| | Dynamic | 100% | 97.27% | 99.84% | 88.38% | 64.82% | 43.91% | 68.71% | 76.93% | **6.24%** | **95.80%** |
| | SIG | 97.13% | 97.13% | 79.28% | 90.50% | 33.98% | 91.83% | **4.64%** | 89.36% | 5.13% | **96.33%** |
| | Average | 99.38% | 96.80% | 94.35% | 87.68% | 21.88% | 83.01% | 19.17% | 87.04% | **7.27%** | **95.25%** |
| ImageNet Subset | *None* | 0% | 89.93% | 0% | 83.14% | 0% | 85.49% | 0% | 88.18% | 0% | **88.31%** |
| | BadNets | 100% | 84.41% | 97.70% | 82.81% | 28.59% | 78.52% | 6.32% | 81.26% | **0.94%** | **87.76%** |
| | Trojan | 100% | 85.56% | 96.39% | 80.34% | 6.67% | 76.87% | 15.48% | 80.52% | **1.47%** | **88.19%** |
| | Blend | 99.93% | 86.15% | 99.34% | 81.33% | **19.23%** | 75.83% | 26.47% | 82.39% | 21.42% | **85.12%** |
| | SIG | 98.60% | 86.02% | 78.82% | 85.72% | 25.14% | 78.87% | 5.15% | 83.01% | **0.18%** | **86.42%** |
| | Average | 99.63% | 85.53% | 93.06% | 82.55% | 19.91% | 77.52% | 13.35% | 81.80% | **6.00%** | **86.87%** |

average ASR against the 10 attacks on CIFAR-10, GTSRB, and the ImageNet subset, respectively. This superiority becomes more significant when compared to other baseline methods.

We notice that our ABL is not always the best when looking at the 10 attacks individually. For instance, NAD is the best defense against the Blend attack on CIFAR-10 and against the SIG attack on GTSRB, while MCR is the best against Blend on GTSRB and the ImageNet subset. We suspect this is because both Blend and SIG mingle the trigger pattern (i.e., another image or superimposed sinusoidal signal) with the background of the poisoned images, producing an effect of natural artifacts. This makes them harder to be isolated and unlearned, since even clean data can have such patterns [24]. This is one limitation of our ABL that needs to be improved in future works. Our ABL achieves a much better performance than the baselines on defending against the 4 feature-space attacks. For example, NAD only manages to decrease the attack success rates of the FC and the DFST attacks to 58.68% and 35.21%, respectively. In contrast, our ABL can bring their ASRs down to below 10%. The Dynamic and the CBA attacks are found to be the toughest attacks to defend against in general. For example, baseline methods NAD, MCR and FP can only decrease CBA's ASR to 33.11%, 38.76%, and 77.33% on CIFAR-10, and Dynamic's ASR to 68.71%, 64.82%, and 99.84% on GTSRB, respectively, a result that is much worse than the 29.81% and 6.24% ASRs of our ABL.

Maintaining the clean accuracy is as important as reducing the ASR, as the model would lose utility if its clean accuracy is much compromised by the defense. By inspecting the average CA results in Table 1, one can find that our ABL achieves nearly the same clean accuracy as models trained on 100% clean (shown in row *None* and column 'No Defense') datasets. Specifically, our ABL surpasses the average clean accuracy of NAD by 4.39% (84.76% vs. 80.37%), 8.21% (95.25% vs. 87.04%) and 5.07% (86.87% vs. 81.80%) on CIFAR-10, GTSRB, and the ImageNet subset, respectively. FP defense decreases the model's performance even when training data are clean (the *None* row). This makes our ABL defense more practical for industrial applications where performance is equally

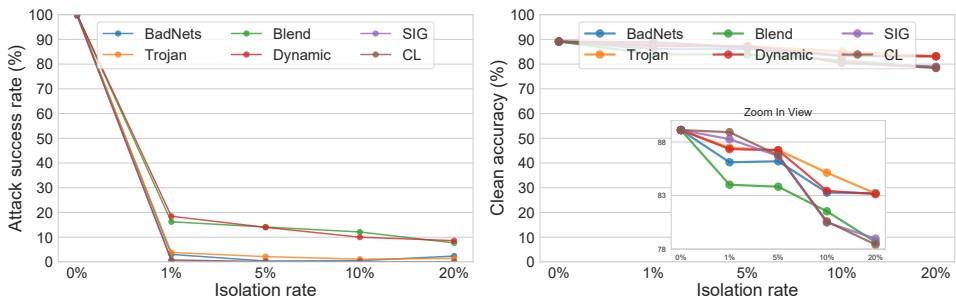

Figure 3: Performance of our ABL with different isolation rate $p \in [0.01, 0.2]$ on CIFAR-10 dataset. Left: attack success rate (ASR); Right: clean accuracy of ABL against 6 classic backdoor attacks.

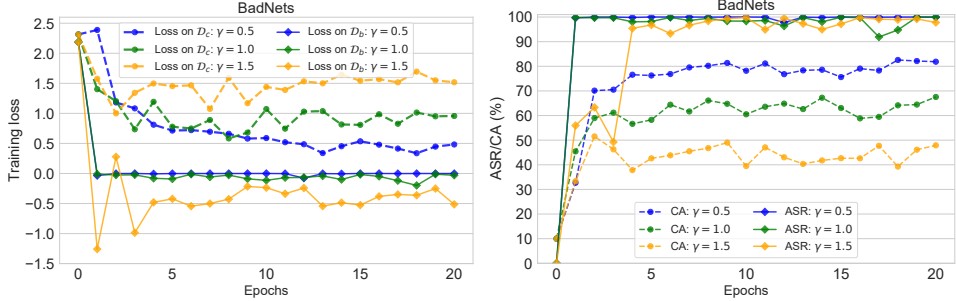

Figure 4: Separation effect of local gradient ascent with different $\gamma$ on CIFAR-10 against BadNets. Left: Training loss on the ground truth backdoor ($\mathcal{D}_b$) and clean ($\mathcal{D}_c$) subsets; Right: Attack success rate (ASR) and clean accuracy (CA). The gap between the two lines of the same color becomes wider for larger $\gamma$, i.e., better separation effect.

important as security. Results of our ABL with 1% isolation against 1% poisoning can be found in Table 10 (Appendix B.9).

**Effectiveness with Different Isolation Rates.** Here, we study the correlation between the isolation rate $p = |\widehat{\mathcal{D}}_b|/|\mathcal{D}|$ and the performance of our ABL, on the CIFAR-10 dataset. We run ABL with different $p \in [0.01, 0.2]$ and show attack success rates and clean accuracies in Figure 3. There is a trade-off between ASR reduction and clean accuracy. Specifically, a high isolation rate can isolate more backdoor examples for the later stage of unlearning, producing a much lower ASR. However, it also puts more examples into the unlearning mode, which harms the clean accuracy. In general, ABL with an isolation rate $< 5\%$ works reasonably well against all 6 attacks, even though the backdoor poisoning rate is much higher, i.e., 70% (see Figure 2 in Section 4.2). Along with the results in Table 1, this confirms that it is indeed possible to break and unlearn the backdoor correlation with only a tiny subset of correctly-identified backdoor examples, highlighting one unique advantage of backdoor isolation and unlearning approaches.

**Effectiveness with Different Turning Epochs.** Here, we study the impact of the timing to switch from the learning stage ($\mathcal{L}_{\text{LGA}}$) to the unlearning stage ($\mathcal{L}_{\text{GGA}}$) on CIFAR-10. We compare four different turning epochs: the 10th, 20th, 30th, and 40th epoch, and record the results of our ABL in Table 6 (see Appendix B.4). We find that delayed turning epochs tend to slightly hinder the defense performance. Despite the slight variations, all choices of the turning epoch help mitigate backdoor attacks, but epoch 20 (i.e., at 20% - 30% of the entire training progress) achieves the best overall results. This trend is consistent on other datasets as well. We attribute these results to the success of LGA in preserving the difference between clean and backdoor samples over time, which enables us to select the turning epoch flexibly. A more comprehensive discussion on LGA is given in Section 4.2.

### 4.2 Comprehensive Understanding of ABL

**Importance of Local Gradient Ascent.** To help understand how LGA works in isolating backdoor data, we visualize and compare in Figure 4 the training loss and the model's performances (ASR and CA) under three different settings where $\gamma$ is set to 0.5, 1.0, and 1.5, respectively. It is evident that LGA can segregate backdoor examples from clean examples to a certain extent under all three settings of $\gamma$ by preventing the loss of clean examples from converging. Moreover, a larger $\gamma$ leads to

Table 2: Stress testing with poisoning rate up to 50% and 70% for 4 attacks including BadNets, Trojan, Blend, and Dynamic on CIFAR10 dataset.

| Poisoning Rate | Defense | BadNets | | Trojan | | Blend | | Dynamic | |
|---|---|---|---|---|---|---|---|---|---|
| | | ASR | ACC | ASR | ACC | ASR | ACC | ASR | ACC |
| 50% | *None* | 100% | 75.31% | 100% | 70.44% | 100% | 69.49% | 100% | 66.15% |
| | ABL | 4.98% | 70.52% | 16.11% | 68.56% | 27.28% | 64.19% | 25.74% | 61.32% |
| 70% | *None* | 100% | 74.8% | 100% | 69.46% | 100% | 67.32% | 100% | 66.15% |
| | ABL | 5.02% | 70.11% | 29.29% | 68.79% | 62.28% | 64.43% | 69.36% | 62.09% |

a wider difference in training loss as well as ASR and CA. However, we note that this may cause training instability, as evidenced by the relatively larger fluctuations with $\gamma = 1.5$.

We also examine the precision of the 1% isolated backdoor set under different $\gamma$ of 0, 0.5, 1.0, and 1.5 on CIFAR-10, GTSRB, and the ImageNet subset. We use BadNets attack with a poisoning rate 10% and set the turning (isolation) epoch of ABL to 20. We report the isolation precision results in Table 7 (see Appendix B.5). As can be seen, when $\gamma = 0$, the detection precision is poor; this indicates that it is tough for the model to tell apart backdoor examples from the clean ones without the LGA, which is foreseeable because the clean training loss is uncontrolled and overlaps with the backdoor training loss. Note that as soon as we set $\gamma > 0$, the precision immediately improves on both CIFAR-10 and the ImageNet subset. Additionally, the precision of the isolation task is not sensitive to the change in $\gamma$, which again allows the hyperparameter value to be flexibly chosen. In our experiments, $\gamma = 0.5$ works reasonably well across different datasets and models.

In summary, LGD creates and sustains a gap between the training loss of clean and backdoor examples, which plays a vital role in extracting an isolated backdoor set.

**Stress Testing: Fixing 1% Isolation Rate While Increasing Poisoning Rate.** Now that we know we can confidently extract a tiny subset of backdoor examples with high purity, the challenge remains whether the extracted set is sufficient for the model to unlearn the backdoor. We demonstrate that our ABL is a stronger method, even under this strenuous setting. Here, we experiment on CIFAR-10 against 4 attacks including BadNets, Trojan, Blend, and Dynamic with poisoning rates up to 50%/70% and show the results in Table 2. We can find that even with a high poisoning rate of 50%, our ABL method can still reduce the ASR from 100% to 4.98%, 16.11%, 27.28%, and 25.74% for BadNets, Trojan, Blend, and Dynamic, respectively. Note that ABL will break when the poisoning rate reaches 70%. In this case however, the dataset should not be used to train any models in the first place. Overall, ABL remains effective against up to 1) 70% BadNets; and 2) 50% Trojan, Blend, and Dynamic. This finding is very compelling, considering ABL needs to isolate only 1% of training data. As we mentioned before, this is because the correlation between the backdoor pattern and the target label exposes a weakness of backdoor attacks. Our ABL utilizes the GGA to break this link and achieve defense goals effortlessly.

### 4.3 Exploring Alternative Isolation and Unlearning Methods

**Alternative Isolation Methods.** In this section, we compare the isolation precision of our ABL with two backdoor detection methods, namely Activation Clustering (AC) [7] and Spectral Signature Analysis (SSA) [6]. The goal is to isolate 1% of training examples into the backdoor set ($\widehat{\mathcal{D}}_b$), and we provide in Figure 7 (see Appendix B.3) the precision of these methods alongside our ABL in detecting the 6 backdoor attacks on CIFAR-10 dataset. We find that both AC and SS achieve high detection rates on BadNets and Trojan attacks while perform poorly on 4 other attacks. A reasonable explanation is that attacks covering the whole image with complex triggers (e.g., Blend, Dynamic, SIG, and CL) give confusing and unidentifiable output representations of either feature activation or spectral signature, making these detection methods ineffective. It is worth mentioning that our ABL is effective against all backdoor attacks with the highest average detection rate. In addition, we find that the flooding loss [38] proposed for mitigating overfitting is also very effective for backdoor isolation. We also explore a confidence-based isolation with label smoothing (LS), which unfortunately fails on most attacks. More details of these explorations can be found in Figure 8 and 9 in Appendix B.6.

**Alternative Unlearning Methods.** Here we explore several other empirical strategies, including image-based, label-based, model-based approaches, to rebuild a clean model on the poisoned data.

Table 3: Performance of various unlearning methods against BadNets attack on CIFAR-10.

| Backdoor Unlearning Methods | Method Type | Discard $\widehat{\mathcal{D}}_b$ | Backdoored | | After Unlearning | |
|---|---|---|---|---|---|---|
| | | | ASR | CA | ASR | CA |
| Pixel Noise | Image-based | No | 100% | 85.43% | 57.54% | 82.33% |
| Grad Noise | Image-based | No | 100% | 85.43% | **47.65%** | **82.62**% |
| Label Shuffling | Label-based | No | 100% | 85.43% | 30.23% | 83.76% |
| Label Uniform | Label-based | No | 100% | 85.43% | 75.12% | 83.47% |
| Label Smoothing | Label-based | No | 100% | 85.43% | 99.80% | 83.17% |
| Self-Learning | Label-based | No | 100% | 85.43% | **21.26%** | **84.38**% |
| Finetuning All Layers | Model-based | Yes | 100% | 85.43% | 99.12% | 83.64% |
| Finetuning Last Layers | Model-based | Yes | 100% | 85.43% | 22.33% | 77.65% |
| Finetuning ImageNet Model | Model-based | Yes | 100% | 85.43% | 12.18% | 75.10% |
| Re-training from Scratch | Model-based | Yes | 100% | 85.43% | 11.21% | 86.02% |
| **ABL** | Model-based | No | 100% | 85.43% | **3.04**% | **86.11**% |

These approaches are motivated by the second weakness of backdoor attacks, and are all designed to break the connection between the trigger pattern and the target class. We experiment on CIFAR-10 with BadNets (10% poisoning rate), and fix the backdoor isolation method to our ABL with a high isolation rate 20% (as most of them will fail with 1% isolation). Table 3 summarizes our explorations. Our core findings can be summarized as: **a)** adding perturbations to pixels or gradients is not effective; **b)** changing the labels of isolated examples is mildly effective; **c)** finetuning some (not all) layers of the model cannot effectively mitigate backdoor attacks; **d)** "self-learning" and "retraining the model from scratch" on the isolated clean set are good choices against backdoor attacks; and **e)** our ABL presents the best unlearning performance. Details of these methods are given in Appendix A.4. The performance of these methods under the 1% isolation rate is also reported in Table 8 in Appendix B.7. We also considered another widely used attack settings that the user only allowed to access a backdoored model and hold limited benign data. In this case, we proposed to combine our ABL unlearning with Neural Cleanse [4] to erase the backdoroed model (see Appendix B.10).

## 5    Conclusion

In this work, we identified two inherent characteristics of backdoor attacks as their weaknesses: 1) backdoor examples are easier and faster to learn than clean examples, and 2) backdoor learning establishes a strong correlation between backdoor examples and the target label. Based on these two findings, we proposed a novel framework - Anti-Backdoor Learning (ABL) - which consists of two stages of learning utilizing local gradient ascent (LGA) and global gradient ascent (GGA), respectively. At the early learning stage, we use LGA to intentionally maximize the training loss gap between clean and backdoor examples to isolate out the backdoored data via the low loss value. We use GGA to unlearn the backdoored model with the isolated backdoor data at the last learning stage. Empirical results demonstrate that our ABL is resilient to various experimental settings and can effectively defend against 10 state-of-the-art backdoor attacks. Our work introduces a simple but very effective ABL method for industries to train backdoor-free models on real-world datasets, and opens up an interesting research direction for robust and secure machine learning.

## Broader Impact

Large-scale data have been key to the success of deep learning. However, it is hard to guarantee the quality and purity of the training data in many cases, and even high-quality datasets may contain backdoors, especially those collected from the internet. By introducing the concept of *anti-backdoor learning* (ABL), our work opens up a new direction for secure and robust learning with not-fully-trusted data. Even in the clean setting, ABL can prevent deep learning models from overfitting to those overly easy samples. Beyond backdoor defense, ABL could be explored as a generic *data-quality-ware* learning mechanism in place of the traditional data-quality-agnostic learning. Such a mechanism may help reduce many potential data-quality-related risks such as memorization, overfitting, backdoors and biases. Although not our initial intention, our work may adversely be exploited to develop advanced backdoor attacks. This essentially requires new defenses to combat.

## Acknowledgement

This work is supported by China National Science Foundation under grant number 62072356 and in part by the Key Research and Development Program of Shaanxi under Grant 2019ZDLGY12-08.

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
