# A Implementation Details

## A.1 Datasets and Classifiers

The datasets and DNN models used in our experiments are summarized in Table 4.

Table 4: Detailed information of the datasets and classifiers used in our experiments.

| Dataset | Labels | Input Size | Training Images | Classifier |
|---------|--------|------------|-----------------|------------|
| CIFAR-10 | 10 | 32 x 32 x 3 | 50000 | WideResNet-16-1 |
| GTSRB | 43 | 32 x 32 x 3 | 39252 | WideResNet-16-1 |
| ImageNet subset | 12 | 224 x 224 x 3 | 12406 | ResNet-34 |

## A.2 Attack Details

We trained backdoored model for 100 epochs using Stochastic Gradient Descent (SGD) with an initial learning rate of 0.1 on CIFAR-10 and the ImageNet subset (0.01 on GTSRB), a weight decay of $10^{-4}$, and a momentum of 0.9. The learning rate was divided by 10 at the 20th and the 70th epochs. The target labels of backdoor attacks were set to 0 for CIFAR-10 and ImageNet, and 1 for GTSRB. Note that the implementation of the Dynamic attack proposed in the original paper is different from the traditional settings of data poisoning. We used their pre-trained generator model to create a poisoned training dataset to train the backdoored model on. We did not use any data augmentation techniques to avoid side-effects on the ASR. The details of backdoor triggers are summarized in Table 5.

Table 5: Attack settings of 6 backdoor attacks. ASR: attack success rate; CA: clean accuracy.

| Attacks | Trigger Type | Trigger Pattern | Target Label | Poisoning Rate |
|---------|--------------|-----------------|--------------|----------------|
| BadNets | Fixed | Grid | 0, 1 | 10% |
| Trojan | Fixed | Reversed Watermark | 0, 1 | 10% |
| Blend | Fixed | Random Pixel | 0, 1 | 10% |
| Dynamic | Varied | Mask Generator | 0, 1 | 10% |
| SIG | Fixed | Sinusoidal Signal | 0, 1 | 10% |
| CL | Fixed | Grid and PGD Noise | 0, 1 | 10% |
| FC | Fixed | Optimization-based | source 1, target 0 | 10% |
| DFST | Fixed | Style Generator | 0 | 10% |
| LBA | Fixed | Optimization-based | 0 | 10% |
| CBA | Varied | Mixer Construction | 0 | 10% |

## A.3 Defense Details

For Fine-pruning (FP)[*], we pruned the last convolutional layer of the model until the CA of the network became lower than that of the other defense baselines. For model connectivity repair (MCR)[†], we trained the loss curve for 100 epochs using the backdoored model as an endpoint and evaluated the defense performance of the model on the loss curve. We adopted the open-source code[‡] used in NAD and finetuned the backdoored student network for 10 epochs with 5% of clean data. The distillation parameter $\beta$ for CIFAR-10 was set to be identical to the value given in the original paper. We cautiously selected the $\beta$ value for GTSRB and ImageNet to achieve the best trade-off erasing results between ASR and CA. All these defense methods were trained using the same data augmentation techniques, i.e., random crop ($padding = 4$), horizontal flipping, and Cutout (1 patch with 9 length).

For our ABL defense, we trained the model for 20 epochs with a learning rate of 0.1 on CIFAR-10 and ImageNet subset (0.01 on GTSRB) before the turning epoch. After isolating 1% of potential backdoor examples, we further trained the model for 60 epochs on the full training dataset (this helps recover the model's clean accuracy), and in the last 20 epochs, we trained the model using the $\mathcal{L}_{GGA}$ loss with the 1% isolated backdoor examples and a learning rate of 0.0001. Note that for ABL, the

---

[*]https://github.com/kangliucn/Fine-pruning-defense
[†]https://github.com/IBM/model-sanitization/tree/master/backdoor/backdoor-cifar
[‡]https://github.com/bboylyg/NAD

data augmentations are only used for the mid-stage of 60 epochs, which can help improve clean accuracy.

All experiments were run on a hardware equipped with a RTX 3080 GPU and an i7 9700K CPU.

### A.4 Details of Alternative Isolation and Unlearning Methods

For the two explored isolation methods, i.e., the flooding loss and the label smoothing, we set the flooding level to 0.5 and the smoothing value to 0.2 and 0.4, respectively. The explored unlearning methods are defined as follows:

- **Pixel Noise**. This method randomly adds Gaussian noise to $\widehat{\mathcal{D}}_b$ then trains the model on the resulting dataset $\widehat{\mathcal{D}}_b^* \cup \widehat{\mathcal{D}}_c$.

- **Grad Noise**. This method adds Gaussian noise to $\widehat{\mathcal{D}}_b$ at the quarter quantile with the largest gradient then trains the model on the resulting dataset $\widehat{\mathcal{D}}_b^* \cup \widehat{\mathcal{D}}_c$.

- **Label Shuffling**. This method applies a random permutation on the labels of examples from $\widehat{\mathcal{D}}_b$ then trains the model on the resulting dataset $\widehat{\mathcal{D}}_b^* \cup \widehat{\mathcal{D}}_c$.

- **Label Uniform**. This method corrupts the labels in $\widehat{\mathcal{D}}_b$ with an uniform random class then trains the model on the resulting dataset $\widehat{\mathcal{D}}_b^* \cup \widehat{\mathcal{D}}_c$.

- **Label Smoothing**. This method decreases the confidence of the original one-hot labels in the $\widehat{\mathcal{D}}_b$ then trains the model on the resulting dataset $\widehat{\mathcal{D}}_b^* \cup \widehat{\mathcal{D}}_c$.

- **Self-learning**. This method relabels $\widehat{\mathcal{D}}_b$ with the model trained from scratch on $\widehat{\mathcal{D}}_c$, then trains the model on the resulting dataset $\widehat{\mathcal{D}}_b^* \cup \widehat{\mathcal{D}}_c$.

- **Finetuning All Layers**. This method finetunes all layers of the backdoored model on $\widehat{\mathcal{D}}_c$.

- **Finetuning Last Layers**. This method finetunes the last flatten layer of the backdoored model on $\widehat{\mathcal{D}}_c$.

- **Finetuning ImageNet Model**. This method finetunes the last block of a pre-trained ImageNet model (i.e., Resnet-34) on $\widehat{\mathcal{D}}_c$.

- **Re-training from Scratch**. This method retrains a model from scratch on $\widehat{\mathcal{D}}_c$.

### A.5 Examples of Backdoor Triggers

Figure 5 shows some backdoor examples used in our experiments.

## B   More Experimental Results

### B.1 Training Loss under Different Poisoning Rates

Figure 6 shows the training loss for the BadNets attack under three different poisoning rates, i.e., 1%, 5%, and 10%. It is evident that the higher the poisoning rate, the faster the training loss declines on backdoor examples.

### B.2 Training Loss on More Datasets

Figure 10 show the results of the training loss on both clean and backdoor examples on GTSRB and ImageNet subset.

### B.3 Comparison of training data detection

We compared our ABL to two state-of-the-art backdoor data detection methods: Activation Cluster (AC) and Spectral Signature Analysis (SSA). We reproduce these two methods using the open source

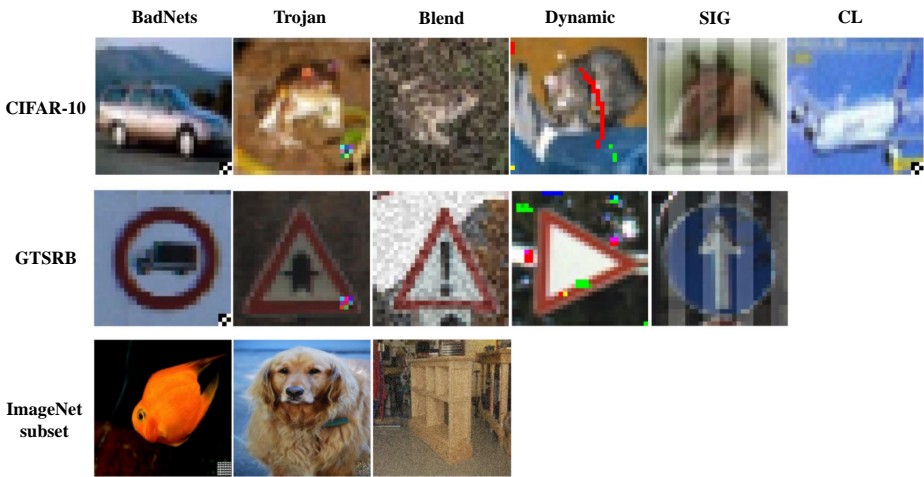

Figure 5: Backdoored images by different attacks for CIFAR-10, GTSRB, and ImageNet.

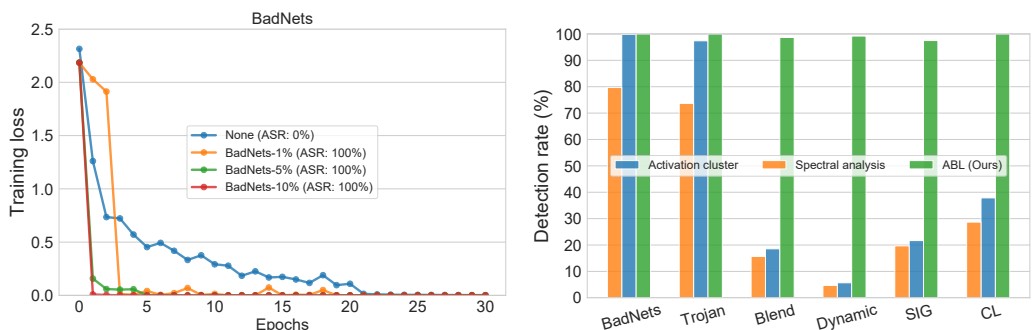

Figure 6: Training loss of BadNets with poisoning rates of 1%, 5%, and 10% on CIFAR-10.

Figure 7: Detection precision ($TP/(TP + FP)$) of the 1% isolated backdoor examples by our ABL and two other state-of-the-art backdoor detection methods: Activation Cluster (AC) and Spectral Signature Analysis (SSA).

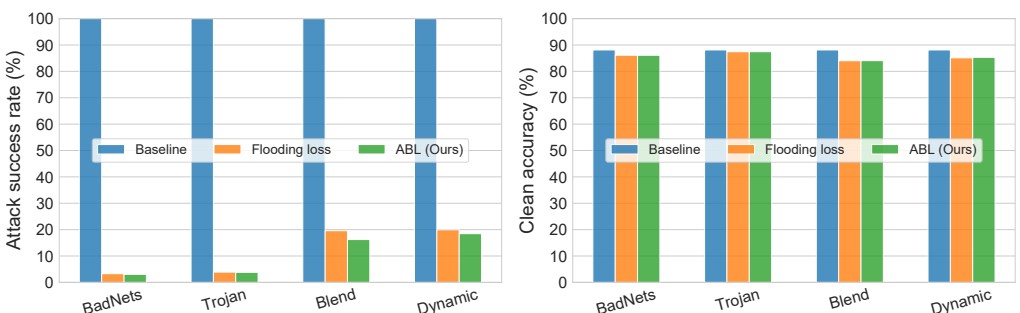

Figure 8: Performance of flooding loss based isolation against 4 backdoor attacks on CIFAR-10.

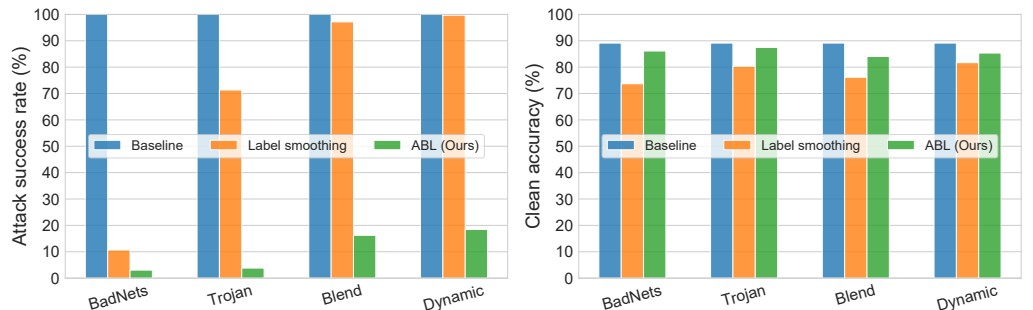

Figure 9: Performance of label smoothing based isolation against 4 backdoor attacks on CIFAR-10.

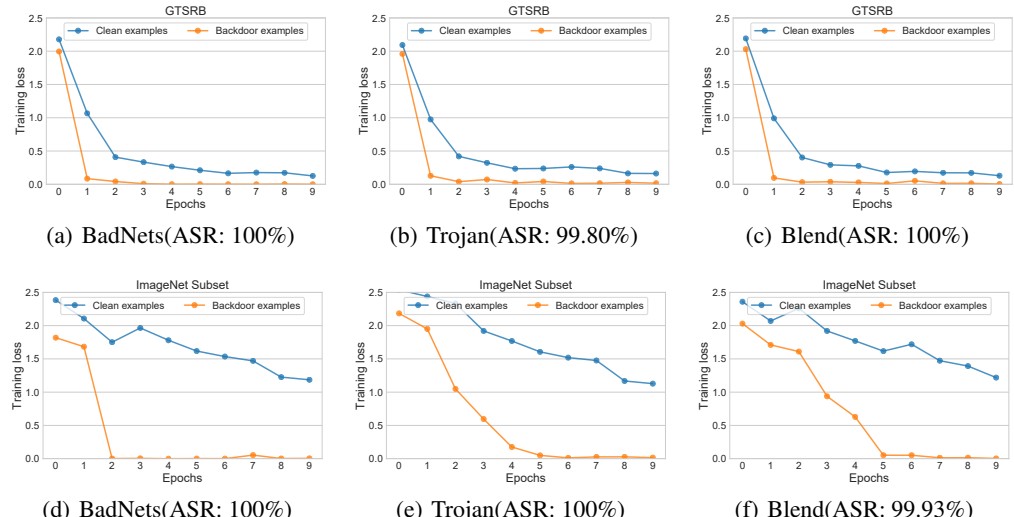

(a) BadNets(ASR: 100%)  (b) Trojan(ASR: 99.80%)  (c) Blend(ASR: 100%)

(d) BadNets(ASR: 100%)  (e) Trojan(ASR: 100%)  (f) Blend(ASR: 99.93%)

Figure 10: The training loss of ResNet models on GTSRB (top row) and the ImageNet subset (bottom row) under a poisoning rate 10%. WideResNet-16-1/ResNet-34 are used for GTSRB/the ImageNet subset, respectively. ASR: attack success rate. Here, we only tested 3 classic attacks: BadNets, Trojan, and Blend.

code§ following the default settings suggested in their papers. As can be seen in Figure 7, our ABL defense achieves the best detection precision against all 6 backdoor attacks on the CIFAR-10 dataset.

## B.4 Results of Tuning Epochs

Table 6 shows the performance of our ABL under four different turning epochs: the 10th, the 20th, the 30th, and the 40th. The best turning epoch is epoch 20 (20% - 30% of the entire training process) with the best defense results.

Table 6: Performance of our ABL with different tuning epochs on CIFAR-10. The isolation rate for ABL is set to 1% while the poisoning rate of the 4 attacks is 10%.

| Tuning Epoch | BadNets | | Trojan | | Blend | | Dynamic | |
|---|---|---|---|---|---|---|---|---|
| | ASR | CA | ASR | CA | ASR | CA | ASR | CA |
| 10 | **1.12%** | 85.30% | 5.04% | 85.12% | 16.34% | **84.22%** | 25.33% | 84.12% |
| **20** | 3.04% | **86.11%** | **3.66%** | **87.46%** | **16.23%** | 84.06% | **18.46%** | **85.34%** |
| 30 | 3.22% | 85.60% | 3.81% | 87.25% | 19.87% | 83.83% | 20.56% | 85.23% |
| 40 | 4.05% | 84.28% | 4.96% | 85.14% | 18.78% | 81.53% | 19.15% | 83.44 |

§https://github.com/ain-soph/trojanzoo

## B.5 Results of Detection Rate under different $\gamma$

Table 7 shows the isolation performances under four different values of $\gamma$. We use BadNets with a poisoning rate 10% as an example attack and run experiments with our ABL on CIFAR-10, GTSRB, and the ImageNet subset. Table 7 shows the precision of the 1% isolation of backdoor examples. The isolation is executed at the end of the 20th training epoch with ($\gamma \geq 0.5$) or without ($\gamma = 0$) our LGA. As the table indicates, the isolation precison is extremely low if LGA is not used. Once we select a $\gamma \geq 0.5$, the precision immediately goes up to 100%, meaning all the isolated backdoor examples are true backdoor examples. Arguably, adaptive attacks could enforce large loss values to circumvent our loss threshold $\gamma$. Until now however, it is not clear in the current literature how to design such attacks without manipulating the training procedure. We will leave this question as future work.

Table 7: Detection precision ($TP/(TP + FP)$) of the 1% isolated examples under different $\gamma$, using BadNets as an example attack.

| Dataset | $\gamma = 0$ | $\gamma = 0.5$ | $\gamma = 1.0$ | $\gamma = 1.5$ |
|---|---|---|---|---|
| CIAFR-10 | 26% | 100% | 100% | 100% |
| GTSRB | 89% | 100% | 100% | 100% |
| ImageNet Subset | 21% | 100% | 100% | 100% |

## B.6 Results of Alternative Backdoor Isolation Methods

The flooding loss [36] is a regularization technique to improve model generalization by avoiding zero training loss. Here, we replace our local gradient ascent (LGA) by the flooding loss to isolate potential backdoored data while fixing the unlearning method to our global gradient ascent (GGA). Note the flooding level is set to 0.5. Figure 8 compares the results of flooding loss based defense to our ABL defense. We find that the two methods achieve a similar performance against 4 backdoor attacks on CIFAR-10. This indicates that the flooding loss is also capable of isolating backdoored data, which may be an unexpected benefit of overfitting-mitigation techniques. Our LGA outperforms the flooding loss against the Blend and the Dynamic attacks in terms of reducing the attack success rate, though only mildly. We would like to point out that LGA serves only one part of our ABL and can potentially be replaced by any backdoor detection methods, and the effectiveness of GLA with 1% of isolated data is also a key to the success of ABL.

As label smoothing (LS) can also alleviate the overconfidence output of the deep networks, we also try to train a model using LS to isolate the examples with higher output confidence levels (often refer to backdoor examples). The comparison results are shown in Figure 9. Unfortunately, we find that LS-based defense achieves much poorer ASR performance against the Dynamic, the Trojan, and the Blend attacks, even with the smoothing value set to 0.4. This might be caused by the similar confidence distribution between clean and backdoor examples.

## B.7 Results of Alternative Backdoor Unlearning Methods

Here, we report the unlearning results of the set of explored unlearning methods under the isolation rate $p = 0.01$ (1%). The results are shown in Table 8. It shows that all these unlearning methods except for our ABL failed to defend against any backdoor attack, with the 100% ASR almost unchanged. This may be caused by the high ratio of backdoored data (9%) remaining in the potential clean set $\widehat{\mathcal{D}}_c$.

## B.8 Results of Computational Complexity for ABL

Here, we report the time cost of the isolation operation of our ABL on CIFAR-10 and the ImageNet subset in Table 9. The additional computational cost is less than 10% and 3% of the standard training time on CIFAR-10 ($\sim$ 40 minutes for 100 epochs) and the ImageNet subset ($\sim$ 80 minutes for 100 epochs), respectively.

Table 8: Performance of various unlearning methods against the BadNets attack on CIFAR-10 under the 1% isolation rate by our ABL.

| Backdoor Unlearning Methods | Method Type | Discard $\widehat{\mathcal{D}}_b$ | Backdoored | | After Unlearning | |
|---|---|---|---|---|---|---|
| | | | ASR | CA | ASR | CA |
| Pixel Noise | Image-based | No | 100% | 85.43% | 100% | 84.72% |
| Grad Noise | Image-based | No | 100% | 85.43% | 100% | 84.63% |
| Label Shuffling | Label-based | No | 100% | 85.43% | 99.98% | 82.76% |
| Label Uniform | Label-based | No | 100% | 85.43% | 99.92% | 83.47% |
| Label Smoothing | Label-based | No | 100% | 85.43% | 100% | 84.71% |
| Self-Learning | Label-based | No | 100% | 85.43% | 100% | 83.91% |
| Finetuning All Layers | Model-based | Yes | 100% | 85.43% | 100% | 85.02% |
| Finetuning Last Layers | Model-based | Yes | 100% | 85.43% | 100% | 67.32% |
| Finetuning ImageNet Model | Model-based | Yes | 100% | 85.43% | 100% | 73.43% |
| Re-traing from Scratch | Model-based | Yes | 100% | 85.43% | 100% | 85.24% |
| ABL (Ours) | Model-based | No | 100% | 85.43% | **3.04%** | **86.11%** |

Table 9: The average time (second) of the isolation operation of ABL on CIFAR10 and the ImageNet subset; CPU: Intel(R) Core(TM) i5-9400F CPU @ 2.90GHz 2.90 GHz); GPU: 1 NVIDIA GeForce RTX 1080 TI.

| Time Cost | CIFAR-10 | ImageNet subset |
|---|---|---|
| | Data Size: 50,000 | Data Size: 12,480 |
| BadNets | ~230 s | ~140 s |
| Trojan | ~228 s | ~140 s |
| Blend | ~232 s | ~138 s |
| SIG | ~228 s | ~136 s |

## B.9  Results of ABL Defense under Low Poisoning Rate (1%)

Table 10 shows the results of ABL with 1% isolated data against the 1% poisoning rate on CIFAR-10 with WRN-16-1. Compared to the 10% poisoning results in Table 1, ABL achieved more ASR reduction with similar clean ACC against 1% poisoning, as expected.

Table 10: ABL unlearning with 1% isolated data against 1% poisoning rate on CIFAR-10.

| Poisoning Rate 1% | BadNets | | Trojan | | Blend | | Dynamic | | SIG | | CL | |
|---|---|---|---|---|---|---|---|---|---|---|---|---|
| | ASR | ACC | ASR | ACC | ASR | ACC | ASR | ACC | ASR | ACC | ASR | ACC |
| Baseline | 99.52 | 85.56 | 97.11 | 83.46 | 99.48 | 83.32 | 99.87 | 82.15 | 62.13 | 84.01 | 54.39 | 83.78 |
| ABL (Ours) | 3.01 | 88.13 | 3.16 | 87.56 | 8.58 | 84.43 | 13.36 | 85.09 | 0.01 | 88.78 | 0 | 88.54 |

Table 11: ABL can help unlearn the backdoors from backdoored models on CIFAR-10. ABL unlearning was applied on 500 trigger patterns reverse-engineered by Neural Cleanse (NC) based on 500 clean training images.

| Defense | BadNets | | Trojan | | Blend | | SIG | |
|---|---|---|---|---|---|---|---|---|
| | ASR | ACC | ASR | ACC | ASR | ACC | ASR | ACC |
| Baseline | 100 | 85.43 | 100 | 82.14 | 100 | 84.51 | 100 | 84.16 |
| NC + ABL Unlearning (Ours) | 0.12 | 85.38 | 5.33 | 79.78 | 1.17 | 83.11 | 3.21 | 80.53 |

## B.10  ABL Unlearning Combined with Neural Cleanse

The threat models of backdoor attacks are mainly classified into three types: a) poisoning the training data [1, 11, 23], b) poisoning the training data and manipulating the training procedure [18, 34, 31], or c) directly modifying parameters of the final model [23]. Our threat model refers to the threat model a), which is one of the widely accepted threat models for backdoor attacks. Different defense settings benefit different types of users (defenders). Particularly, defenses developed under our threat model could benefit companies, research institutes, or government agencies who have the resources

to train their own models but rely on outsourced training data. It also benefits MLaaS (Machine Learning as a Service) providers such as Amazon ML and SageMaker, Microsoft Azure AI Platform, Google AI Platform and IBM Watson Machine Learning to help users train backdoor-free machine learning models. Note that our focus is the traditional machine learning paradigm with a single dataset and model. Backdoor attacks on federate learning (FL) follow a different setting thus require different defense strategies [42, 43].

Here, we show that our ABL method can also help other defense settings, where the defender can only purify a backdoored model with a small subset of clean data (e.g., only 1% clean training data is available). In this case, ABL can leverage existing trigger pattern detection methods like Neural Cleanse [4] to reverse engineer a set of trigger patterns, then unlearn the backdoor from the model via its maximization term (defined on the reverse-engineered trigger patterns and their predicted labels). Table 11 shows the effectiveness of this simple approach. ABL can effectively and effortlessly unlearn the trigger from a backdoored model with the reversed trigger patterns. This demonstrates the usefulness of our ABL in purifying backdoored models, closing the gap between backdoor detection and backdoor erasing.

### B.11  Visualization of the Isolated Backdoor Examples

Figure 11 and Figure 12 show a few examples of those isolated backdoor images under the BadNets attack on CIFAR-10, with or without our ABL isolation.

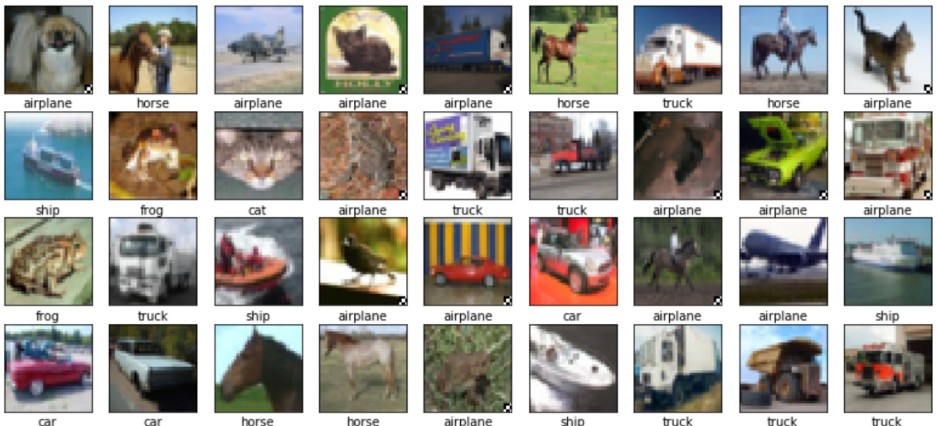

Figure 11: Backdoor images isolated without our ABL ($\gamma = 0$) under the BadNets attack on CIFAR-10.

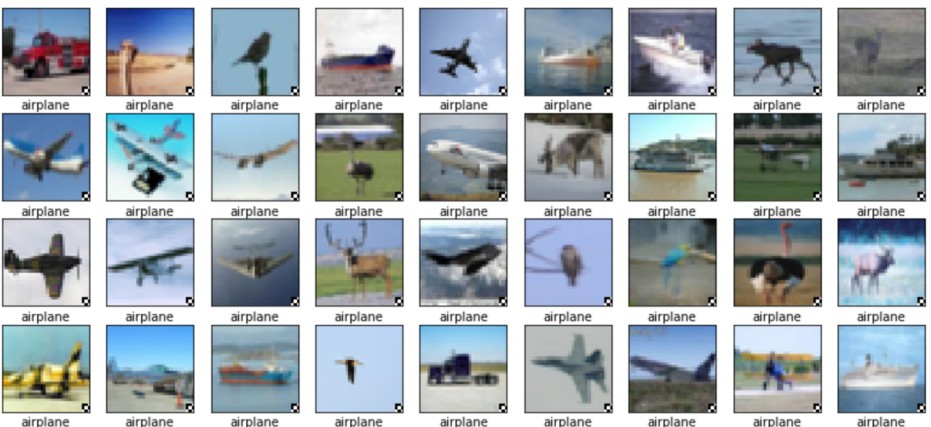

Figure 12: Backdoor images isolated with our ABL ($\gamma = 0.5$) under the BadNets attack on CIFAR-10.