# OpenReview forum: "Anti-Backdoor Learning: Training Clean Models on Poisoned Data"
_NeurIPS.cc/2021/Conference — NeurIPS 2021 Poster_

### Official Review · Reviewer_oQMj · 2021-07-12

**Rating:** 6
**Confidence:** 4

**Summary:**

This paper observes that models converge much faster on backdoored data than clean data. Within a few epochs, the training loss for backdoored data will decrease to zero, but not for clean data. Based on such an observation, this paper aims to train clean models on poisoned data without knowing the backdoored portion in the training set. Specifically, it proposes a new training procedure called anti-backdoor learning (ABL), which splits the traditional training procedure into two stages. In the first stage, ABL uses a threshold to prevent loss values for individual samples being too small by flipping the sign of the loss. It then selects a small portion (1%) of the training set with small losses as the backdoored set and the remaining as the clean set. In the second stage, ABL applies gradient ascent on the backdoored set and gradient descent on the clean set. The evaluation is conducted on three standard benchmarks and ABL is compared with three existing backdoor removal baselines. The experimental results show ABL has the best performance on both reducing attack success rate and retaining normal accuracy in most cases. A list of studies on different components of ABL and comparisons with other alternative methods are interesting and informative.

**Limitations And Societal Impact:**

The authors have discussed the limitations of ABL against Blend and SIG attacks. But there are missing discussions on the applicability of ABL on label-specific attacks, filter-like attacks. Please see details in the main review.

The authors have addressed the potential negative societal impact.

**Main Review:**

The observation on the training loss of backdoored data is interesting. The proposed training procedure leverages this observation to counter backdoor effects, and the results show the effectiveness of the proposed technique. This work may have potentials in defending against backdoor attacks in general. I have a few concerns regarding the observation, the proposed method and experiments. It would be appreciated if the authors could elaborate in the following points.

1. The observation of the distinctive train losses between backdoored data and clean data is quite interesting. But the six attacks studied in Figure 2 all seem to be universal attacks, where the backdoor flips all other classes to one single target class. How about label-specific attacks where only one or two victim classes are compromised? Can we still have similar observations for the training loss as in Figure 2? If so, the proposed method would probably be much more general and useful. Also, 5 out of 6 studied attacks are static attacks. Although Dynamic attack stamps input-specific triggers, the type of triggers is still at the pixel-level, where such patterns might easily be learned by models. This could be the reason of the distinctive train losses. To further understand different types of backdoors, filter attacks would be a good candidate for the study as such attacks universally transform the input images and generate more natural backdoored data. It would be interesting to see whether such natural-looking backdoored data are still distinguishable from clean data based on their training losses.

2. From Figure 2, by just looking at the training loss for the first a few epochs, backdoored data are already clearly separated from clean data. Why does ABL only select a small portion (1%) as the backdoored data? Why cannot we just directly use a threshold, for instance 0.4, to remove all the backdoored data? In Figure 2, the 0.4 threshold at epoch 5 can clearly separate the two sets (backdoored and clean). Are there some poisoned images still having large training loss after a few epochs or clean images having very small loss? If this is the case, Figure 2 does not well reflect the true training loss. Could the authors clarify this?

3. The proposed local gradient ascent loss is very similar to the flooding loss. The results in Figure 9 in Appendix shows these two also have very similar performance. Since the evaluation is only conducted on 4 attacks on CIFAR-10, it is hard to tell whether the proposed loss is indeed better than the flooding loss. Also, the flooding level is set to 0.5 as stated in the appendix. Is this the best parameter for the flooding loss? Could there be a better parameter? I would suggest the authors to extend the evaluation on the comparison with the flooding loss as the proposed loss has a very similar characteristic and performance.

4. In the introduction, the authors motivate their work by claiming "a high attack success rate can still be achieved on CIFAR-10 by different attacks even though the poisoning rate is less than 1%". This is very intriguing as such a low poisoning rate might make defense harder. However, in the evaluation, this paper uses 10% poisoning rate as the default setting. Even when studying the performance of ABL against different poisoning rates, this paper only increases the poisoning rate. I am interested in whether ABL is still effective if a very low poisoning rate is chosen. I would like to see ABL's performance on the poisoning rates studied in Figure 1.

5. Have the authors considered adaptive attacks? For instance, the adversary can intentionally generate poisoned data having training loss similar to normal data. Will the proposed training procedure still be effective defending against backdoor attacks?

6. Could the authors explain why ABL has a higher clean accuracy on the clean-label poisoned training set (89.03%) than on the clean training set (88.41%)?

**Time Spent Reviewing:**

6

---

> ### Author Response · Authors · 2021-08-10
> **Response to Reviewer oQMj - Part Ⅰ**
>
> Thanks for the accurate summarization of our work and the valuable comments. Please find our response below.
>
> ---
> **Q1:** The six attacks studied in Figure 2 all seem to be universal attacks, where the backdoor flips all other classes to one single target class. How about label-specific attacks where only one or two victim classes are compromised? Can we still have similar observations for the training loss as in Figure 2?
>
> **A1:** Thanks for the insightful comment. Out of the 6 tested attacks, 4 (i.e., BadNets, Trojan, Blend, and Dynamic) are *dirty-label* attacks that flip the labels of other class samples to the target class, while the rest of the 2 (i.e., SIG and CL) attacks are *clean-label* attacks which only poison the target class sample without flipping the labels. The 2 clean-label attacks can be regarded as the suggested “only one victim class” case. We are not sure if restricting the 4 *dirty-label* attacks to fewer classes (e.g., 2 or 4) will decrease their strength, but are happy to add such an experiment to the appendix. We believe the conclusion will be the same as choosing different poisoning samples does not change the nature of the attack. By the way, the following new results demonstrate the same *easy learning* phenomenon of a more complicated Feature-collision attack [1] (suggested by other reviewers), and our ABL can also defend this attack. We hope this can help clarify your concern.
>
> (`The training loss of Feature-collision attack`)
>
> |              Epochs              |   1    |   2    |   3    |   4    |   5    |   6    |   7    |   8    |   9    |   10   |
> | :------------------------------: | :----: | :----: | :----: | :----: | :----: | :----: | :----: | :----: | :----: | :----: |
> |   Clean examples training loss   | 1.2174 | 1.2041 | 1.0154 | 1.0538 | 1.0702 | 0.9715 | 0.8663 | 0.9523 | 0.8822 | 0.8179 |
> | Backdoor examples  training loss | 1.0731 | 1.0688 | 0.6982 | 0.5162 | 0.4983 | 0.4092 | 0.6767 | 0.6424 | 0.3855 | 0.3129 |
>
> (`ABL Defense against  Feature-collision attack under different turning epochs. `)
>
> | ABL Defense | Turning Epoch | Attack Baseline | 10 | 20 | 30 | 40 |
> |:---:|:---:|:---:|:---:|:---:|:---:|-----:|
> | Feature-collision | ASR/ACC(%) | 88.52 /  83.32 | 11.18 / 82.63 | 0.14 / 83.85 | 0.08 / 84.36 | 0.15 / 83.11 |
>
> ---
> **Q2:** To further understand different types of backdoors, filter attacks would be a good candidate for the study as such attacks universally transform the input images and generate more natural backdoored data. It would be interesting to see whether such natural-looking backdoored data are still distinguishable from clean data based on their training losses.
> **A2:** Thanks for the suggestion. We were unable to test the suggested filter attacks, which we assume are the  Nashville and Gotham filter attacks used in [2], as we couldn’t be able to find the code for generating the two attacks. We will check with the authors and are happy to add such an experiment if the code is available.
>
> ---
> **Q3:** From Figure 2, by just looking at the training loss for the first a few epochs, backdoored data are already clearly separated from clean data. Why does ABL only select a small portion (1%) as the backdoored data? Why cannot we just directly use a threshold, for instance 0.4, to remove all the backdoored data? In Figure 2, the 0.4 threshold at epoch 5 can clearly separate the two sets (backdoored and clean). Are there some poisoned images still having large training loss after a few epochs or clean images having very small loss? If this is the case, Figure 2 does not well reflect the true training loss. Could the authors clarify this?
>
> **A3:** Thanks for the insightful question. You are absolutely right that, with a properly chosen threshold, we can easily isolate all backdoored samples. We did not adopt this strategy for two reasons. **First**, in real-world scenarios, it could be hard to determine the threshold and estimate the exact poisoning rate (explained in our threat model at Lines 122-123). So, we decided to free the users from making such a decision by simply restricting the isolation rate to be as small as possible such that removing the isolated data would have minimum impact on the entire learning process whatsoever. It is also to our surprise that 1% isolation and unlearning can be very effective against many attacks. We believe we have indeed successfully uncovered one common weakness of backdoor attacks. **Second**, we want to fully exploit the training data including the poisoned part. For instance, if 70% of the data is poisoned, we believe it is not ideal to discard the 70% of data, conversely, we aim to train a model that is as good as if the whole dataset is clean. This will make our ABL method more useful in real-world scenarios where a large proportion of the training data was collected from multiple untrusted sources.
>
> To answer your other questions, 1) poisoned images generally have much smaller losses after a few epochs of training, and 2) yes, clean samples can also have small loss unless our LGA loss is applied.
>
>
> ---
> **Q4:** The proposed local gradient ascent loss is very similar to the flooding loss. The results in Figure 9 in Appendix shows these two also have very similar performance. Since the evaluation is only conducted on 4 attacks on CIFAR-10, it is hard to tell whether the proposed loss is indeed better than the flooding loss. Also, the flooding level is set to 0.5 as stated in the appendix. Is this the best parameter for the flooding loss? Could there be a better parameter? I would suggest the authors to extend the evaluation on the comparison with the flooding loss as the proposed loss has a very similar characteristic and performance.
>
> **A4:** Thanks for the thoughtful comment. Yes, our LGA loss can also be realized by the flooding loss, as we mentioned at Lines 206-207 and 325. And yes, the flooding parameter of the flooding loss was deliberately tuned and $b=0.5$ was found to be optimal. Our LGA outperforms the flooding loss against Blend and Dynamic attacks in terms of reducing the attack success rate, though only mildly.  We will add the tuning results of the flooding parameter to the appendix. Meanwhile, we would like to point out that LGA serves only one part of our ABL and can potentially be replaced by any backdoor detection methods, and the effectiveness of GLA with 1% isolated data is also key to the success of ABL.
>
> ---
> **Q5:** In the introduction, the authors motivate their work by claiming "a high attack success rate can still be achieved on CIFAR-10 by different attacks even though the poisoning rate is less than 1%". This is very intriguing as such a low poisoning rate might make defense harder. However, in the evaluation, this paper uses 10% poisoning rate as the default setting. Even when studying the performance of ABL against different poisoning rates, this paper only increases the poisoning rate. I am interested in whether ABL is still effective if a very low poisoning rate is chosen. I would like to see ABL's performance on the poisoning rates studied in Figure 1.
>
> **A5**: This is a very fair question. Apologize that we focused too much on the large poisoning rates, which we thought are more challenging to defend. Please find the new results below for the 1% poisoning rate on CIFAR-10 with WRN-16-1. Compared to the 10% poisoning results in Table 1, ABL achieved more ASR reduction with similar clean ACC against 1% poisoning,  as expected.
>
> (`ABL defense under 1% poisoning rate on CIFAR-10`)
>
> | Poisoning 1%  |  | BadNets | Trojan | Blend | Dynamic | SIG | CL |
> |:---:|:---:|:---:|:---:|:---:|:---:|:---:|:---:|
> | Attack Baseline | ASR/  ACC(%) | 99.52/  85.56 | 97.11/ 83.46 | 99.48/ 83.32 | 99.87/ 82.15 | 62.13/ 84.01 | 54.39/ 83.78 |
> | ABL | ASR/  ACC(%) | 3.01/ 88.13 | 3.16/ 87.56 | 8.58/ 84.43 | 13.36/ 85.09 | 0.01/ 88.78 | 0/ 88.54 |

---

> > ### Author Response · Authors · 2021-08-10
> > **Response to Reviewer oQMj - Part Ⅱ**
> >
> > **Q6:** Have the authors considered adaptive attacks? For instance, the adversary can intentionally generate poisoned data having training loss similar to normal data. Will the proposed training procedure still be effective defending against backdoor attacks?
> >
> > **A6:** Thanks for the insightful question. We indeed tried to make the 6 attacks to be more adaptive to ABL by using adversarial perturbations to increase the loss. However, we find this makes these attacks even easier to defend. This is because adversarial perturbations can only increase the **test** loss rather than the **training** loss, that is, high test loss samples are not necessary having high training loss when used to train a model. We are not aware of any existing attacks that can surely increase the training loss of a sample without tampering with the training procedure. We do believe adaptive attacks are possible, but they will be very difficult to design as *natural* patterns are generally not powerful enough to trick the model to **always** predict a target class. We hope it is ok to leave this exploration to our future work.
> >
> > ---
> > **Q7:** Could the authors explain why ABL has a higher clean accuracy on the clean-label poisoned training set (89.03%) than on the clean training set (88.41%)?
> >
> > **A7:** Thank you very much for the insightful observation. We have noticed this as well. We suspect it may be caused by the randomness that exists in independent training, rather than a signal that ABL might help generalization. We believe ABL has the potential to be applied to improve generalization but will need carefully tuned parameters and more in-depth analysis on the turning epoch, isolation rate, and other factors that may impact learning.
> >
> >
> > ---
> > [1] Ali Shafahi, W Ronny Huang, Mahyar Najibi, Octavian Suciu, Christoph Studer, Tudor Du-mitras, and Tom Goldstein.  Poison frogs! targeted clean-label poisoning attacks on neural networks. In NeurIPS, 2018.
> > [2] Liu, Yingqi, et al. "ABS: Scanning neural networks for back-doors by artificial brain stimulation." Proceedings of the 2019 ACM SIGSAC Conference on Computer and Communications Security. 2019.

---

> ### Author Response · Authors · 2021-08-31
> **Thanks to Reviewer oQMj**
>
> We would like to thank the reviewer for the detailed comments, and particularly, for recognizing the potential of our work in defending backdoor attacks in general.
>
> We hope our response has adequately addressed your concerns regarding experiments with more backdoor attacks, why ABL isolates only 1% training data, the similarity of ABL to flooding loss, and its effectiveness with low poisoning rates. Please note that, based on the discussions with other reviewers, by far we have verified the “easy learning” phenomenon on 10 backdoor attacks.
>
> Kindly let us know if anything is unclear. We truly appreciate your valuable feedback and comments that help us further improve our work.

---

> > ### Comment · Reviewer_oQMj · 2021-09-01
> > **Thanks for the Response**
> >
> > I appreciate the authors provide the response with additional experiments.
> >
> > The response addresses most of my questions and concerns. I still have a few comments.
> >
> > For the label-specific attack, I was talking about BadNets and Trojan those dirty-label attacks with one victim label and one target label. This is different from the universal attack with one target label and others as victim labels. If the authors can include related experiments in the paper, that would be great. Otherwise, I would suggest to make it clear about the treat model for these attacks in the paper.
> >
> > For the filter attack, the implementation of Nashville and Gotham filters are not very complex and can be found online. I believe the poisoning procedure is similar to those dirty-label attacks. It would be great if the authors can try them out. If this will not be included in the paper, the authors may want to add a discussion or clarity this.
> >
> > Lastly, the adaptive attack can be designed by generating those poisoned inputs while maximizing their losses in the first few epochs and minimizing their losses in the later epochs. Basically, it is to make the learning of poisoned samples similar to normal samples. Since the authors were not able to empirically evaluate on the adaptive attack, it would be better to at least discuss it in the paper.
> >
> > I am keeping my initial score.

---

> > > ### Author Response · Authors · 2021-09-02
> > > **Thanks**
> > >
> > > Thank you very much for the additional comments. We can definitely add these expriments.

---

### Official Review · Reviewer_KE43 · 2021-07-16

**Rating:** 7
**Confidence:** 4

**Summary:**

Authors have introduced the concept of anti-backdoor learning to learn clean models using backdoor-poisoned data. They have proposed Anti-Backdoor Learning (ABL) to automatically break backdoor attack during training with poisoned data. They have identified two weaknesses from backdoor attack. First weakness is that models learn from poisoned data at a much faster rate than these learn from clean data. The stronger the attack, the faster the models converge to a backdoor. The second weakness is that the backdoor task is tied to a specific class (the backdoor target class). Based on the two weaknesses, the authors designed a two-step method to learn a clean model with poisoned data.  The first step is to isolate poisoned samples from the training dataset, and the second step is to use the isolated samples to unlearn backdoor. The first step happened in the early training stage and the second stage takes over the training after the first stage. As compared to other state of the arts, the isolation step is very precise, and in many case, the precision is easily to reach 100%. The second step is very effective to unlearn backdoors. Even if only 1% of the poisoned samples are isolated, it can counter up to 70% poisoned samples to learn a clean model. Overall, this is a good paper and the proposed method outperformed the competing methods under the authors’ experimental setting.

**Ethical Concerns:**

None.

**Limitations And Societal Impact:**

In addition to the limitations mentioned by the authors, here are some other concerns of this reviewer: all the showcases are based on the relatively simple attack: BadNets. For example, The Stress Testing showed that with a fixing 1% Isolation rate, the proposed method is still very effective with up to poising rate of 70%, which is fantastic meaning that the 1% identified poisoned samples can overpower the other unidentified 69% poisoned samples with the proposed method. Is it true for other relatively complicated attacks?

**Main Review:**

Originality: Are the tasks or methods new? Is the work a novel combination of well-known techniques? (This can be valuable!) Is it clear how this work differs from previous contributions? Is related work adequately cited?

This is a new method as long as I am aware of, and I think references are adequate.

Quality: Is the submission technically sound? Are claims well supported (e.g., by theoretical analysis or experimental results)? Are the methods used appropriate? Is this a complete piece of work or work in progress? Are the authors careful and honest about evaluating both the strengths and weaknesses of their work?

The paper is technically sound. Claims are well supported by their experimental results. However, the models being used in the paper are way below the performances of the state of the arts. For example, state of the arts on CIFAR10 are way above 90% (close to 100%), and the models being used in this paper obtained less than 90% of accuracies. How would this affect your conclusions?

Clarity: Is the submission clearly written? Is it well organized? (If not, please make constructive suggestions for improving its clarity.) Does it adequately inform the reader? (Note that a superbly written paper provides enough information for an expert reader to reproduce its results.)

The paper is well organized, pleasant to read and easy to follow.

Significance: Are the results important? Are others (researchers or practitioners) likely to use the ideas or build on them? Does the submission address a difficult task in a better way than previous work? Does it advance the state of the art in a demonstrable way? Does it provide unique data, unique conclusions about existing data, or a unique theoretical or experimental approach?

I believe the solution provided by the paper is significant. However, the authors did not discuss the computational complexities. For example, if the user gives a pass of poisoned sample detection before using the data to train a model, how much extra time s/he needs to spend, assuming the poisoned data screening process is effective? This information will give important information of the importance of the paper.


**Time Spent Reviewing:**

10 hours

---

> ### Author Response · Authors · 2021-08-10
> **Response to Reviewer KE43**
>
> Thanks for the valuable feedback. Please find our response below.
>
> ---
> **Q1:** The models being used in the paper are way below the performances of the state of the arts. For example, state of the arts on CIFAR10 are way above 90% (close to 100%), and the models being used in this paper obtained less than 90% of accuracies. How would this affect your conclusions?
>
> **A1** Thanks for the question. For a fair comparison, we have closely followed previous works to choose the target models, attacks, and their training/attacking settings. Although we didn’t deliberately tune the target models to match the SOTA, we believe the conclusions will be the same as our ABL is not restricted to a particular model nor training setting. Please note that, although a clean acc close to 100% can be achieved by NAS, EfficientNet, or transformers, the WideResNet-16-1 we adopted for CIFAR-10 can only achieve 92%-94% clean accuracy with advanced data augmentations like AutoAugment.  We will add one example analysis with SOAT results on CIFAR-10 to the appendix.
>
>
> ---
> **Q2:** Computational complexity of poisoned data screening (backdoor isolation).
>
> **A2:** Thanks for the thoughtful comment. Please find the time cost of the isolation operation of our ABL on CIFAR-10 and the ImageNet subset in the table below. The additional computational cost is less than 10% and 3% of the standard training time on CIFAR-10 (~ 40 minutes for 100 epochs) and ImageNet subset (~ 80 minutes for 100 epochs), respectively.
>
> (`The average time (second) of the isolation operation of ABL on CIFAR10 and ImageNet subset; CPU: Intel(R) Core(TM) i5-9400F CPU @ 2.90GHz   2.90 GHz); GPU: 1 NVIDIA GeForce RTX 1080 TI.`)
>
> | Time Cost | CIFAR-10 (data size: 50,000) | ImageNet subset (data size: 12,480) |
> |:---:|:---:|:---:|
> | BadNets | ~ 230 s | ~ 140 s |
> | Trojan | ~ 228 s | ~ 140 s |
> | Blend | ~ 232 s | ~ 138 s |
> | SIG | ~ 228 s | ~ 136 s |
>
>
> ---
>
> **Q3:** The Stress Testing showed that with a fixing 1% Isolation rate, the proposed method is still very effective with up to poising rate of 70%, which is fantastic meaning that the 1% identified poisoned samples can overpower the other unidentified 69% poisoned samples with the proposed method. Is it true for other relatively complicated attacks?.
>
> **A3:** Yes, ABL is extremely effective against other attacks as well. Please find the performance of 1% ABL against 70% and 50% poisoning of all attacks in the table below. Note that, when the poisoning rate increases, the clean acc will drop for all models due to corruption of the training labels. Overall, ABL remains effective against up to 1) 70% BadNets; and 2) 50% Trojan, Blend, and Dynamic. We believe this is an exciting result for backdoor defense, considering ABL only isolates 1% data. We will update the “Stress Testing” section with the new results and adjust our claims.
>
> (`Stress testing with poisoning rate up to 50% and 70% on CIFAR10. We didn’t report the results of SIG and CL as they are clean label attacks whose maximum poisoning rate is only 10% (i.e., poisoning the entire class) of the entire training set.`)
>
> | Poisoning rate 70%  |  | BadNets | Trojan | Blend | Dynamic | SIG | CL |
> |:---:|:---:|:---:|:---:|:---:|:---:|:---:| :---:|
> | Attack Baseline | ASR/ACC(%) | 100/74.8 | 100/69.46 | 100/67.32 | 100/66.15 | - | - |
> | ABL | ASR/ACC(%) | 5.02/70.11 | 29.29/68.79 | 62.28/64.43 | 69.36/62.09 | - | - |
>
>
> | Poisoning rate 50%  |  | BadNets | Trojan | Blend | Dynamic | SIG | CL |
> |:---:|:---:|:---:|:---:|:---:|:---:|:---:|-----|
> | Attack Baseline | ASR/ACC(%) | 100/75.31 |100/70.44 | 100/69.49 | 100/66.15 | - | - |
> | ABL | ASR/ACC(%) | 4.98/70.52| 16.11/68.56 | 27.28/64.19 | 25.74/61.32 | - | - |

---

> ### Author Response · Authors · 2021-08-31
> **Thanks to Reviewer KE43**
>
> We would like to thank the reviewer for taking the time to review our paper and the positive feedback, and in particular for recognizing the significance of our finding and the proposed method.
>
> Kindly let us know whether we have adequately addressed your comments on the performance of the baseline models and the time cost of ABL. We truly appreciate your valuable comments which help us improve the clarity of our experimental setting and our claim regarding the stress testing.

---

> > ### Comment · Reviewer_KE43 · 2021-09-01
> > **Keep my original score.**
> >
> > I appreciate the authors' feedbacks and extra experiment results that sufficiently addressed my comments. Based on the discussions among authors and other reviewers, and the extra experiments, I would like to keep my original score.

---

### Official Review · Reviewer_9SWb · 2021-07-16

**Rating:** 5
**Confidence:** 4

**Summary:**

This paper observes that deep neural networks learn backdoored data faster than
benign samples. Based on this finding, the paper proposes Anti-Backdoor Learning
(ABL), which can learn a benign model on poisoned datasets. Specifically, at the
beginning of model training, ABL maximizes the training loss gap between clean
samples and backdoored examples and detects backdoored data based on the
differences in their loss values. Then, ABL unlearns backdoored model with the
detected data.

**Limitations And Societal Impact:**

The work uses existing assets but did not discuss license issues.

**Main Review:**


This paper finds that deep neural models learn backdoored samples faster than
benign samples, which is interesting. However, it lacks sufficient analysis to
support this conclusion. For example, the paper only shows the results on
ResNet-18 and CIFAR-10 in figure 2 to support the conclusion. One model and one
data set are not representative enough.

Also, the observation is drawn from simple backdoor attacks. I suggest training
them on complicated backdoor attacks, such as feature space backdoor and latent
space backdoor, e.g., Cheng et al. AAAI 2021, Lin et al. CCS 2020, Yao et al.
CCS 2019.

Moreover, the paper assumes that data poisoning happens at the beginning of the
model training. In the data poisoning attack scenario, backdoor attacks can
happen at any time. If the attacker poisons the data dynamically, how would this
defense effectively defend such attacks?

It is unclear to me how this approach can generalize to other threat models,
such as federated learning.

Finally, it seems the selection of the fixed loss value is quite important and
hard to decide. If the fixed loss value is large, the training loss of the model
cannot decrease after it reaches the fixed loss value and the accuracy of the
model cannot increase. If the fixed loss value is small, it is harder to detect
backdoored samples. Based on my understanding, the fixed loss value needs to be
adjusted based on the validation dataset but it is hard to create a validation
dataset for poisoned datasets because we do not know whether the attack exists
and the attack type. The paper can add a discussion about this problem.

A minor comment:

In Table 1, it would be helpful if the paper can provide results of the SIG
attack on ImageNet subset dataset because it seems ABL may perform worse when
defense against SIG attack (from the results on GTSRB).

In "Effectiveness with Different Turning Epochs", the paper claims that epoch 20
achieves the best results for all datasets but only shows results on CIFAR-10.
It would be better if the paper can show results on other datasets.

Updates after rebuttal:

Thank the authors for providing new experiments and justify their design/arguments. They addressed most of my concerns. I would like to raise my score but I still have a concern that when the trigger is complex enough (i.e., close to benign features learned by neural networks), the model will take longer time to learn, in which case, the foundation of this defense is broken.

**Time Spent Reviewing:**

2

---

> ### Author Response · Authors · 2021-08-10
> **Response to Reviewer 9SWb - Part Ⅰ**
>
> We thank the reviewer for the thoughtful comments. We provide our responses and additional evaluations below to address the concerns.
>
> ---
> **Q1:** The results in Figure 2 with one single model (ResNet-18) on one single dataset (CIFAR-10) are not enough to support the conclusion.
>
> **A1:** Thanks for pointing this out. We can definitely add more results with ResNet-34 and the ImageNet subset to the revision. As we are not able to show plots here, we hope the following numerical results can help clarify your concern. Meanwhile, we would like to point out that the effectiveness of ABL across different models and datasets in our experiments can also support our findings.
>
>
> (`Training loss of BadNets, Trojan and Blend attacks on GTSRB dataset`)
>
> | Attacks | Epoch | 1 | 2 | 3 | 4 | 5 | 6 | 7 | 8 | 9 | 10 |
> |:---:|:---:|:---:|:---:|:---:|:---:|:---:|:---:|:---:|:---:|:---:|:---:|
> | BadNets | Clean | 2.178 | 1.066 | 0.408 | 0.333 | 0.266 | 0.211 | 0.165 | 0.176 | 0.173 | 0.126 |
> | - | Backdoor | 0.196 | 0.085 | 0.042 | 0.009 | 0.001 | 0.002 | 0.001 | 0.000 | 0.003 | 0.000 |
> | Trojan | Clean | 2.093 | 0.975 | 0.420 | 0.323 | 0.234 | 0.238 | 0.261 | 0.239 | 0.164 | 0.162 |
> | - | Backdoor | 0.960 | 0.128 | 0.039 | 0.072 | 0.020 | 0.042 | 0.013 | 0.016 | 0.028 | 0.016 |
> | Blend | Clean | 2.092 | 0.991 | 0.403 | 0.292 | 0.277 | 0.177 | 0.194 | 0.173 | 0.172 | 0.129 |
> | - | Backdoor | 0.531 | 0.096 | 0.032 | 0.038 | 0.028 | 0.012 | 0.052 | 0.014 | 0.015 | 0.004 |
>
>
> (`Training loss of BadNets and SIG on ImageNet subset`)
>
> | Attacks | Epoch | 1 | 2 | 3 | 4 | 5 | 6 | 7 | 8 | 9 | 10 |
> |:---:|:---:|:---:|:---:|:---:|:---:|:---:|:---:|:---:|:---:|:---:|:---:|
> | BadNets | Clean | 2.383 | 2.106 | 1.751 | 2.765 | 1.980 | 1.118 | 2.235 | 1.470 | 1.226 | 1.186 |
> | - | Backdoor | 1.819 | 1.683 | 0.002 | 0.005 | 0.000 | 0.002 | 0.001 | 0.054 | 0.003 | 0.005 |
> | SIG | Clean | 2.391 | 1.909 | 1.462 | 1.760 | 1.370 | 1.265 | 1.312 | 1.122 | 0.992 | 1.062 |
> | - | Backdoor | 1.960 | 0.128 | 1.041 | 0.906 | 0.591 | 0.442 | 0.095 | 0.114 | 0.306 | 0.160 |
>
>
> ---
> **Q2:** Also, the observation is drawn from simple backdoor attacks. I suggest training them on complicated backdoor attacks, such as feature space backdoor and latent space backdoor, e.g., Cheng et al. AAAI 2021 [1], Lin et al. CCS 2020 [2], Yao et al. CCS 2019 [3].
>
> **A2:**  Thanks for the suggestion. Please find the new results in the 4 tables below for the suggested Lin et al.  CCS 2020 attack [2] and the feature-collision attack [4] suggested by *Reviewer tvph*. The 1st table shows the *easy learning* phenomenon of the suggested two attacks;  the 2nd table reports the detection precision of the 1% isolated data; table 3 reports the defense performance of our ABL; table 4 provides a side-by-side comparison between our ABL and the best baseline defense NAD. The Lin et al.  CCS 2020 attack is indeed more difficult to isolate and defend against compared with the feature-collision attack and other evaluated attacks.  However, the isolation precision is still very high (> 70%) and ASR was reduced by our ABL from 90.63% to 29.81% (using the default turning epoch 20) with almost no loss of the clean accuracy. This result again outperforms the best baseline NAD.  We will add this analysis to the revision.
>
> We hope these new results are useful and are happy to run more experiments should you have more suggestions. This discussion is extremely valuable for improving our work.
>
> (`We have reproduced two more complicated backdoor attacks: Lin et al. CCS 2020, and feature space backdoor -- Feature-collision attack. The training losses are shown below.`)
>
> | Attacks |  Epochs  |   1    |   2    |   3    |   4    |   5    |   6    |   7    |   8    |   9    |   10   |
> | :---: | :---: | :---: | :---: | :---: | :---: | :---: | :---: | :---: | :---: | :---: | :---: |
> |Feature- collision |   Clean   | 1.2174 | 1.2041 | 1.0154 | 1.0538 | 1.0702 | 0.9715 | 0.8663 | 0.9523 | 0.8822 | 0.8179 |
> | - | Backdoor | 1.0731 | 1.0688 | 0.6982 | 0.5162 | 0.4983 | 0.4092 | 0.6767 | 0.6424 | 0.3855 | 0.3129 |
> |    Lin et al. CCS 2020   |  Clean   | 1.440 | 1.391 | 1.090 | 0.885 | 0.758 | 0.797 | 0.681 | 0.732 | 0.692 | 0.616 |
> |    -    | Backdoor | 1.390 | 1.038 | 1.041 | 0.806| 0.6391 | 0.622 | 0.525 | 0.514 | 0.536 | 0.460 |
>
>
> (`Detection rate of Isolating 1\% data under different turning epochs`)
>
> |  | Turning Epoch | 10 | 20 | 30 | 40 |
> |:---:|:---:|:---:|:---:|:---:|:---:|
> | Detection precision (Isolation 1\%) | Feature-collision | 87.2% | 99.2% | 99.4% | 97.4% |
> | Detection precision (Isolation 1\%) | Lin et al.  CCS 2020 | 73.52% | 75.34% | 71.15% | 73.33% |
>
> (`ABL Defense against Feature- collision and Lin et al.  CCS 2020 attacks under different turning epochs.` )
>
> | ABL Defense | Turning Epoch | Attack Baseline | 10 | 20 | 30 | 40 |
> |:---:|:---:|:---:|:---:|:---:|:---:|---:|
> | Feature-collision | ASR/ACC(%) | 88.52 /  83.32 | 11.18 / 82.63 | 0.14 / 83.85 | 0.08 / 84.36 | 0.15 / 83.11 |
> | Lin et al.  CCS 2020 | ASR/ACC(%) | 90.63 /  84.72 | 30.48 / 84.15 | 29.81 / 84.66 | 35.56 / 84.78| 30.11 / 84.47 |
>
>
> (`Our ABL Defense V.S NAD Defense`)
>
> | Defense |  | Attack Baseline | NAD | ABL(Ours) |
> |:---:|:---:|:---:|:---:|:---:|
> | Feature-collision | ASR/ACC(%) | 88.52/ 83.32 | 58.68/ 81.23 | **0.08/ 84.36** |
> | Lin et al.  CCS 2020 | ASR/ACC(%) | 90.63/ 84.72 | 35.21/ 82.40 | **29.81/ 84.66** |
>
>
> ---
> **Q3:**  Moreover, the paper assumes that data poisoning happens at the beginning of the model training. In the data poisoning attack scenario, backdoor attacks can happen at any time. If the attacker poisons the data dynamically, how would this defense effectively defend such attacks?
>
>
> **A3:** Thanks for the thoughtful comment. Our work adopts the standard backdoor threat model that has been widely used in previous works like [5-8], that is, the adversary cannot tamper with the training process nor manipulating the final model. This is because, otherwise, the backdoor trigger will be assumed to be surely implanted into the target model and there is nothing we can do to avoid it other than purifying the model via a postprocessing step. While we agree that other threat models are also realistic especially in open-source model sharing where the models are periodically updated, we would like to argue that different threat models generally require different types of methods to defend. As we explained in the paper, the focus of our work is to explore “Is it possible to train a clean model on poisoned data”, a new and challenging research problem we believe is of great practical value for the industry to train backdoor-free models on crowd-sourced data.
>
>
> ---
> **Q4:** It is unclear to me how this approach can generalize to other threat models, such as federated learning.
>
> **A4:** Thank you for raising such an interesting question. While the ABL method itself cannot be directly applied to defend FL backdoors, we believe our finding “backdoor correlation is easier to learn and unlearn” and the proposed ABL mechanism (isolate -> unlearn) are transferable to FL. With on intention to complicate this discussion, please allow us to share some of our findings in a preliminary exploration. We find that the simpler correlation enforced by backdoor attacks exhibits unique characteristics in the uploaded gradients and, by adaptively pruning (set to zeros) or reversing (to the previous communication round) the associated dimensions of the gradient can effectively remove the backdoor during the FL process. However, we would like to point out that this is out of the scope of our work, which is to protect traditional machine learning models from backdoors. We genuinely hope we are not penalized for not conducting the FL experiments, which we believe is an interesting future work.
>
> ---
> **Q5:** The selection of the fixed loss value is quite important and hard to decide.
>
> **A5:** Thanks for the question. The loss threshold $\gamma$ should be determined based on the attack (i.e. the poisoned data) rather than the clean training data. Since most of the backdoor attacks are strong, the poisoned data can easily reach an extremely low loss value (the loss of clean samples can be either low or high depending on the dataset and the sample). This is why a fixed $\gamma=0.5$ works reasonably well across different datasets in our evaluation. Arguably, adaptive attacks could enforce large loss values to circumvent our defense. We tried one such adaptive attack using adversarial perturbations but found that it can still be defended by ABL easily. While we believe more advanced attacks are possible, it is not clear in the current literature how to design such attacks without manipulating the training procedure. We will add the analysis to our revision and are happy to hear some advice from the reviewer.
>
> ---
> **Q6:** In Table 1, it would be helpful if the paper can provide results of the SIG attack on ImageNet subset dataset because it seems ABL may perform worse when defense against SIG attack (from the results on GTSRB).
>
> **A6:** While we believe this is somewhat not a fair evaluation of SIG since it was not proposed nor evaluated for ImageNet in the original paper, we have run the suggested experiment on the ImageNet subset. The result is reported in the table below where it shows ABL remains effective for SIG on the ImageNet subset.
>
> (`ABL defense against SIG on the Imagenet subset`)
>
> |  | ASR(%) | CA(%) |
> |:---:|:---:|:---:|
> |Attack Baseline  | 98.60 | 86.02 |
> | ABL(Ours) | **0.18** | **86.42** |

---

> > ### Author Response · Authors · 2021-08-10
> > **Response to Reviewer 9SWb - Part Ⅱ**
> >
> > **Q7:** In "Effectiveness with Different Turning Epochs", the paper claims that epoch 20 achieves the best results for all datasets but only shows results on CIFAR-10. It would be better if the paper can show results on other datasets.
> >
> > **A7:** Thanks for the suggestion. Please find the new results on the ImageNet subset below. Apparently, epoch 20 is not optimal on ImageNet, although it is already very effective in terms of both attack success rate (ASR) and clean accuracy (ACC). Setting the turning epoch to 30 or 40 could further improve our results. The turning epoch should be selected based on the duration of the “early training stage” on a given dataset, which can be “20% - 30% of the entire training progress” as we suggested at Line 276.
> >
> > (`ABL defense against BadNets and SIG attacks on ImagNet subset under different turning epochs`)
> >
> > | ABL Defense | Turning Epoch | Attack Baseline | 10 | 20 | 30 | 40 |
> > |:---:|:---:|:---:|:---:|:---:|:---:|-----|
> > | BadNets | ASR/    ACC(%) | 100 / 84.41 | 1.18 / 87.63 | 0.94 / 87.76 | 0.88 / 87.36 | 0.65 / 87.11 |
> > | SIG | ASR/    ACC(%) | 98.60 / 86.02 | 2.24 / 85.01 | 1.76 / 86.13 | 0.18 / 86.42| 0.17 / 86.33 |
> >
> > - We are happy to provide more clarifications and run more experiments if there is anything still not clear or should you have other suggestions.
> >
> > ---
> > [1] Cheng et al. Deep Feature Space Trojan Attack of Neural Networks by Controlled Detoxification. AAAI, 2021.
> > [2] Lin et al. Composite Backdoor Attack for Deep Neural Network by Mixing Existing Benign Features. CCS 2020.
> > [3] Yao et al. Latent Backdoor Attacks on Deep Neural Networks. CCS 2019.
> > [4] Shafahi et al.  Poison frogs! targeted clean-label poisoning attacks on neural networks. In NeurIPS, 2018.
> > [5] Gu et al, Badnets: Identifying vulnerabilities in the machine learning model supply chain. arXiv preprint arXiv:1708.06733, 2017.
> > [6] Chen et al, Targeted backdoor attacks on deep learning systems using data poisoning. arXiv preprint arXiv:1712.05526, 2017.
> > [7] Liu  et al, Trojaning attack on neural networks. In NDSS, 2018.
> > [8] Anh Nguyen and Anh Tran. Input-aware dynamic backdoor attack. In NeurIPS, 2020

---

> ### Author Response · Authors · 2021-08-25
> **A follow up message**
>
> Dear Reviewer 9SWb, Thanks again for the valuable comments. Please let us know if anything is unclear. We truly appreciate this opportunity to improve our work and shall be most grateful for any feedback you could give to us.

---

> > ### Comment · Reviewer_9SWb · 2021-08-25
> > **Follow up questions**
> >
> > Dear authors,
> >
> > Thank you very much for the new results and thoughtful comments. I still have some follow up questions.
> >
> > For Q2, thank you for trying this on Lin et al. CCS 2020. Do you have similar observations on Cheng et al. AAAI 2021 and Yao et al. CCS 2019?
> >
> > For the threat model, most poisoning based backdoor attacks assume the supply chain attack model, e.g., BadNets, TrojanNN. In this case, I do not think the authors' understanding that "the adversary cannot tamper with the training process nor manipulating the final model" is correct. The adversary in supply chain attacks usually has full control for a short time. To make sure that my understanding is correct, I got confirmations from the original paper authors that my understanding is correct.
> >
> > In fact, there is a short description of this issue in the IARPA TrojAI project call, which I cite here:
> >
> > ''' Obvious defenses against Trojan attacks include securing the training data (to protect data from manipulation), cleaning the training data (to make sure the training data is accurate), and protecting the integrity of a trained model (prevent further malicious manipulation of a trained clean model). Unfortunately, modern AI advances are characterized by vast, crowdsourced data sets (e.g., 109 data points) that are impractical to clean or monitor. Additionally, many bespoke AIs are created by transfer learning: take an existing, public AI published online and modify it a little for the new use case. Trojans can persist in an AI even after such transfer learning. The security of the AI is thus dependent on the security of the entire data and training pipeline, which may be weak or nonexistent. Furthermore, the user may not be the one doing the training. Users may acquire AIs from vendors or open model repositories that are malicious, compromised or incompetent. Acquiring an AI from elsewhere brings all of the problems with the data pipeline, as well as the possibility of the AI being modified directly while stored at a vendor or in transit to the user. Given the diffuse and unmanageable supply chain security, the focus for the TrojAI Program is on the operational use case where the complete AI is already in the would-be users’ hands: detect if an AI has a Trojan, to determine if it can be safely deployed. ''' (Source: https://www.iarpa.gov/index.php/research-programs/trojai/trojai-baa)
> >
> > Could you further justify this threat model?
> >
> > FL is a typical case where an eval client can decide when to poison the dataset. In such cases, does this approach still work? Could you provide your insights?
> >
> > Thanks,

---

> > > ### Author Response · Authors · 2021-08-27
> > > **Further Clarifications**
> > >
> > > Thanks for your prompt reply! Please allow us to make the following clarifications.
> > >
> > > ---
> > >
> > > **Q1:** For Q2, thank you for trying this on Lin et al. CCS 2020. Do you have similar observations on Cheng et al. AAAI 2021 and Yao et al. CCS 2019?
> > >
> > > **A1:** We observe a similar phenomenon for Cheng et al. AAAI 2021 as indicated by the results in Table 1 below.
> > >
> > > For Yao et al. CCS 2019, we have received the code and the Pubfig dataset from the authors and updated the results in Table 2 below. We also report the result based on our own implementation of Yao et al. CCS 2019 on CIFAR-10. The phenomenon can also be verified on this attack.
> > >
> > > `(Table 1: The training loss of AAAI21 attack on CIFAR-10)`
> > >
> > > |              Epochs              |   1    |   2    |   3    |   4    |   5    |   6    |   7    |   8    |   9    |   10   |
> > > | :------------------------------: | :----: | :----: | :----: | :----: | :----: | :----: | :----: | :----: | :----: | :----: |
> > > |   Clean data training loss   | 1.474 | 1.237 | 1.101 | 0.712 | 0.700 | 0.676 | 0.677 | 0.676 | 0.751 | 0.675 |
> > > | Backdoor data training loss | 2.886 | 0.614 | 0.016 | 0.046 | 0.033 | 0.001 | 0.004 | 0.001 | 0.014 | 0.001 |
> > >
> > >
> > > `(Table 2: The training loss of CCS 19 attack on CIFAR-10.  Note that we also included the training loss of Yao et al. CCS 2019 in the last two rows (marked by Pubfig).)`
> > >
> > > |              Epochs              |   1    |   2    |   3    |   4    |   5    |   6    |   7    |   8    |   9    |   10   |
> > > | :------------------------------: | :----: | :----: | :----: | :----: | :----: | :----: | :----: | :----: | :----: | :----: |
> > > |   Clean data training loss   | 1.979 | 1.656 | 1.360 | 1.017 | 0.8580 | 0.769 |  0.696 | 0.679 | 0.665 | 0.653 |
> > > | Backdoor data training loss | 3.708 | 1.557 | 0.898 | 0.533 | 0.145 | 0.112 | 0.073 | 0.061 | 0.044 | 0.038 |
> > > |   Clean data training loss (Pubfig )   | 11.328 | 8.339 | 7.802 | 7.584 | 7.4760 | 7.427 |  7.380 | 7.359 | 7.330 | 7.311 |
> > > | Backdoor data training loss (Pubfig ) | 13.708 | 9.557 | 5.398 | 4.543 | 3.115 | 2.913 | 3.073 | 2.961 | 2.134 | 2.038 |
> > >
> > > ---
> > >
> > > **Q2:** For the threat model, most poisoning based backdoor attacks assume the supply chain attack model, e.g., BadNets, TrojanNN. In this case, I do not think the authors' understanding that *the adversary cannot tamper with the training process nor manipulate the final model* is correct. The adversary in supply chain attacks usually has full control for a short time. To make sure that my understanding is correct, I got confirmations from the original paper authors that my understanding is correct.
> > >
> > > **A2:** You are absolutely right about the diverse threat models adopted by different attacks: **a)** poisoning the training data [2,3,4,12,13], **b)** poisoning the training data and manipulating the training procedure [1,5,6,7,8], or **c)** directly modifying parameters of the final model [2]. Our threat model refers to the *threat model a)*, which is one of the widely accepted threat models for backdoor attacks. For example, in [12] (ICCV 2021), it says “We assume that attackers are allowed to poison some training data, whereas they have no information on or change other training components (e.g., training loss, training schedule, and model structure)”. Or in [13] (CVPR 2021), it explains “Like existing backdoor attacks [12, 21, 20, 43], we assume the attacker can inject a small number of “dirty label” samples into the training data but has no further control of model training or knowledge of the internal weights and architecture of the trained model.” Threat model a) is also the standard threat model for the broader data poisoning attacks.
> > >
> > > As summarized in [5] (NeurIPS 2020), the corresponding backdoor defenses are **1)** training defense (e.g., detect and remove poisoned samples from the training data [9,10]); **2)** model defense (e.g., backdoor model detection, backdoor removal, model repair); and **3)** test-time defense (e.g., backdoor test sample detection). Our threat model is the same as the threat model adopted by *training defense* methods. For instance, in Spectral Signatures [10] (NeurIPS 2018), it states “We assume the adversary has access to the training data and knowledge of the user’s network architecture and training algorithm, but *does not train the model*. Rather, *the user trains the classifier, but on the possibly corrupted data received from an outside source*.”
> > >
> > > Different defense settings benefit different types of users (defenders). Particularly, defenses developed under threat model a) could benefit companies, research institutes, or government agencies who have the resources to train their own models but rely on outsourced training data. It also benefits MLaaS (Machine Learning as a Service) providers such as Amazon ML and SageMaker, Microsoft Azure AI Platform, Google AI Platform, and IBM Watson Machine Learning to help users train backdoor-free machine learning models.
> > >
> > > While we focus on threat model a), it doesn’t mean our ABL method cannot help other defense settings. For example, given a backdoored model, ABL can leverage existing trigger pattern detection methods like the Neural Cleanse [11] to reverse engineer a set of trigger patterns then unlearn the backdoor from the model via its maximization term (defined on the reverse-engineered trigger patterns and their predicted labels). Table 3 below shows the effectiveness of this simple approach. ABL can effectively and effortlessly unlearn the trigger from a backdoored model with the reversed trigger patterns. This demonstrates the usefulness of our ABL in purifying backdoored models obtained under threat models b) and c).
> > >
> > > `(Table 3: ABL can help unlearn the backdoors from backdoored models on CIFAR-10. ABL unlearning was applied on 500 trigger patterns reverse-engineered by Neural Cleanse(NC) based on 500 clean training images. We have added the new defense results against the original Yao et al. CCS19 (marked by * Pubfig *) to the last row.)  `
> > >
> > > |  Attack |  | Baseline | NC + ABL unlearning (Ours) |
> > > |:---:|:---:|:---:|:---:|
> > > | BadNets | ASR/ACC(%) | 100/ 85.43 | 0.12/ 85.38 |
> > > | Trojan | ASR/ACC(%) | 100/ 82.14 | 5.33/ 79.78 |
> > > | Blend | ASR/ACC(%) | 100/ 84.51 | 1.17/ 83.11 |
> > > | SIG | ASR/ACC(%) | 99.46/ 84.16 | 3.21/ 80.53 |
> > > | Cheng et al. AAAI21  | ASR/ACC(%) | 99.76/ 82.50 | 7.86/ 79.2 |
> > > | Yao et al. CCS19 | ASR/ACC(%) | 99.13/ 81.37 | 0.06/ 80.52 |
> > > | Yao et al. CCS19 (Pubfig) | ASR/ACC(%) | 100/ 97.37 | 5.78/ 95.32 |
> > > ---
> > >
> > > **Q3:** FL is a typical case where an eval client can decide when to poison the dataset. In such cases, does this approach still work? Could you provide your insights?
> > >
> > > **A3:** The above method (i.e., reserve engineering + ABL unlearning) can also be applied to defend FL. In fact, it can be made more convergence-friendly as convergence is a major concern in FL, especially under the Non-IID setting. I.e., the unlearning can be done on *a small number of neurons* (more convergence friendly) that are mostly activated by the reserve engineered trigger patterns. This defense can be applied either locally to the local model to defend one particular participant, or globally at the server side to the global model to defend all participants. Please note that unlearning the global model is less ideal since it will require a tiny subset of clean data at the server side (not sure if public task-irrelevant data can help at this stage). We will add these explorations of FL defense to the appendix.
> > >
> > > ---
> > >
> > > [1] Gu et al. BadNets: Identifying Vulnerabilities in the Machine Learning Model Supply Chain. arXiv.
> > > [2] Liu et al. Trojaning attack on neural networks. NDSS 2018.
> > > [3] Chen et al. Targeted Backdoor Attacks on Deep Learning Systems Using Data Poisoning. arXiv.
> > > [4] Liu et al. Reflection Backdoor: A Natural Backdoor Attack on Deep Neural Networks. ECCV 2020.
> > > [5] Nguyen and Tran. Input-Aware Dynamic Backdoor Attack. NeurIPS 2020.
> > > [6] Cheng et al. Deep Feature Space Trojan Attack of Neural Networks by Controlled Detoxification. AAAI 2021.
> > > [7] Lin et al. Composite Backdoor Attack for Deep Neural Network by Mixing Existing Benign Features. CCS 2020.
> > > [8] Yao et al. Latent Backdoor Attacks on Deep Neural Networks. CCS 2019.
> > > [9] Chen et al. Detecting Backdoor Attacks on Deep Neural Networks by Activation Clustering. AAAI Workshop 2019.
> > > [10] Tran, Li, and Mądry. Spectral signatures in backdoor attacks. NeurIPS 2018.
> > > [11] Wang et al. Neural cleanse: Identifying and mitigating backdoor attacks in neural networks, S&P 2019.
> > > [12] Li et al. Invisible Backdoor Attack with Sample-Specific Triggers. ICCV 2021.
> > > [13] Wenger et al. Backdoor Attacks Against Deep Learning Systems in the Physical World. CVPR 2021.

---

> > > > ### Comment · Reviewer_9SWb · 2021-09-01
> > > > **Comments on clarifications**
> > > >
> > > > Dear authors,
> > > >
> > > > Thank you very much for clarifications and new results. I believe adding such results and analysis will greatly improve the paper.

---

> > > > > ### Author Response · Authors · 2021-09-01
> > > > > **Thanks**
> > > > >
> > > > > Thank you very much for the feedback. We will make sure the new results and analyses are properly incorporated into the revision.

---

> ### Author Response · Authors · 2021-08-30
> **We have received the code of Yao et al. CCS 2019**
>
> Dear Reviewer 9SWb,
> Just to let you know that we have received the code from the authors and will update our result in the previous message once the experiment is completed. Thank you very much for your time.

---

> ### Author Response · Authors · 2021-08-31
> **Thanks to Reviewer 9SWb**
>
> We would like to thank the reviewer for the valuable discussion. We have now included the result of Yao et al. CCS 2019 in our previous response. Both attacks (Cheng et al. AAAI 2021 and  Yao et al. CCS 2019) exhibit the “easy learning” phenomenon. So far, the phenomenon has been confirmed on 10 attacks. We have also included the defense result of a simple trigger reverse engineering plus our ABL unlearning approach against Yao et al. CCS 2019 in Table 3, demonstrating the usefulness of our finding and the ABL method in other backdoor settings. We hope these results can help clarify some of your concerns.

---

### Official Review · Reviewer_tvph · 2021-07-21

**Rating:** 6
**Confidence:** 4

**Summary:**

The proposed work identifies that backdoor attack examples converge fairly quickly compared to clean examples during training, due to the nature of their objective. Using this, the authors identify backdoor examples and use a gradient ascent method to unlearn the backdoor, while gradient descent is used for clean examples. The proposed method is shown to be effective in reducing Attack Success Rate (ASR) while increasing Clean Accuracy (CA) on multiple datasets and attacks.

**Ethical Concerns:**

There are no ethical concerns with this work.

**Limitations And Societal Impact:**

The authors discuss limitations and potential negative impacts.

**Main Review:**

Strengths:

— The proposed work is shown to be effective against multiple attacks for CIFAR10,GTSRB and ImageNet subset dataset.

— The paper is well written and easy to understand.

— The proposed method is compared against different unlearning and detection methods, showing best performance.

Weaknesses:

— While the authors consider different class of attacks, it is not known if the training loss observation holds for other forms of optimization or feature-collision based attacks [15,17,19].

Additional comments:

— In L141, it is mentioned that 6 backdoor attacks are used to poison 10% of CIFAR10 training data. Does that mean the poisoned dataset consists of all 6 attacks? If so, it would be interesting to see if transferability across attacks hold in this scenario- training data consists of 5 backdoor attacks and test set backdoor data comes from the 6th.

Due to the gradient ascent procedure, the network might unlearn only those 5 backdoors. As a result, the Re-training from scratch method (presented in Table 2) might perform better compared to ABL on ASR. An experiment in this setting would help understand the effect of L_GGA.

— How does Tuning epoch affect the weaker SIG attack?

**Time Spent Reviewing:**

7

---

> ### Author Response · Authors · 2021-08-10
> **Response to Reviewer tvph**
>
> Thank you very much for the valuable comments. Please find our responses below.
>
> ---
> **Q1:** It is not known if the training loss observation holds for other forms of optimization or feature-collision based attacks [15,17,19]?
>
> **A1:** This is a fair concern. We have run an additional experiment with the suggested feature-collision attack [1] to confirm the generality of the “easy learning” phenomenon. The results can be found in the table below, where it shows feature attacks also exhibit this phenomenon for the first 10 epochs.
>
> (`The training loss of Feature-collision attack`)
>
> |              Epochs              |   1    |   2    |   3    |   4    |   5    |   6    |   7    |   8    |   9    |   10   |
> | :------------------------------: | :----: | :----: | :----: | :----: | :----: | :----: | :----: | :----: | :----: | :----: |
> |   Clean data training loss   | 1.2174 | 1.2041 | 1.0154 | 1.0538 | 1.0702 | 0.9715 | 0.8663 | 0.9523 | 0.8822 | 0.8179 |
> | Backdoor data  training loss | 1.0731 | 1.0688 | 0.6982 | 0.5162 | 0.4983 | 0.4092 | 0.6767 | 0.6424 | 0.3855 | 0.3129 |
>
>
>
> (`The training loss of UAP optimization attack`)
>
> |              Epochs              |   1    |   2    |   3    |   4    |   5    |   6    |   7    |   8    |   9    |   10   |
> | :------------------------------: | :----: | :----: | :----: | :----: | :----: | :----: | :----: | :----: | :----: | :----: |
> |   Clean data training loss   | 1.2367 | 1.1373 | 0.9854 | 0.9517 | 0.8580 | 0.7349 |  0.6014 | 0.5132 | 0.4587 | 0.4524 |
> | Backdoor data training loss | 0.9575 | 0.7619 | 0.5574 | 0.4728 | 0.3665 | 0.3164 | 0.3507 | 0.2214 | 0.26845 | 0.2433 |
>
>
>
> We agree with the reviewer that backdoor attacks don’t have to manipulate the data, e.g., they may directly modify the training procedure or model parameters once given full access to the training process and the final model, and such modifications may not be reflected in the loss. We haven’t tested this type of attack in our experiments as they adopt a different (stronger) threat model. I.e., it assumes full access to the training data, training procedure, and the final model, which basically is to assume that a backdoor will surely be implanted into the model, and there is nothing we can do to avoid it. Even under this threat model, we believe our finding that “backdoor correlation is simpler and easier to learn and be unlearned” can still help motivate more advanced defenses. We will defer this exploration to future work and restrict our scope in this paper to the standard threat model where the adversary can only manipulate the training data. We believe our robust learning method is of more practical value under this threat model, especially when dealing with crowdsourced but untrusted data in real-world applications.
>
>
> ---
> **Q2:** In L141, it is mentioned that 6 backdoor attacks are used to poison 10% of CIFAR10 training data. Does that mean the poisoned dataset consists of all 6 attacks?
>
> **A2:** “6 backdoor attacks are used to poison 10% of CIFAR10 training data” means the 6 attacks were applied separately to poison CIFAR-10 via 6 sets of independent experiments. In each set of experiments with one particular attack, the poisoning rate was set to 10%. We apologize for the misunderstanding and will make it clearer. We believe the effectiveness of the same ABL mechanism across all 6 sets of experiments has provided sufficient evidence for the “transferability” or “attack-agnostic” property of ABL.
>
> ---
> **Q3:** How does the turning epoch affect the weaker SIG attack?
>
> **A3:** Thanks for the insightful question. The results can be found below. SIG can be removed more effectively when turning epoch >20 is used, as evidenced by the decreasing of ASR. This is because SIG is a weaker attack in the relative sense to other attacks (see Fig. 2), thus requiring more epochs to be learned. Once it is well-learned into the model, it becomes as easy to isolate and remove as other attacks. Note that turning epoch 20 is sufficient to defend SIG and is also the default setting used in our experiments. We didn’t deliberately tune the turning epoch for different types of attacks as the attack type is assumed to be unknown.
>
> (`Turning epoch for SIG attacks on CIFAR-10`)
>
> | Turning Epoch | ASR(\%) | CA(\%) |
> |:---:|:---:|:---:|
> | 10 | 1.51 | 88.24 |
> | 20 | 0.09 | 88.27 |
> | 30 | 0.03 | 88.51 |
> | 40 | 0.02 | 88.23 |
> ---
>
> [1] Ali Shafahi, W Ronny Huang, Mahyar Najibi, Octavian Suciu, Christoph Studer, Tudor Du-mitras, and Tom Goldstein.  Poison frogs! targeted clean-label poisoning attacks on neural networks. In NeurIPS, 2018.

---

> > ### Comment · Reviewer_tvph · 2021-09-01
> > **Thanks for the response**
> >
> > Thanks to the authors for addressing my concerns.  I believe that the additional experiments including feature-collision attacks which have been used widely and the effect of turning epoch can improve understanding of the paper. The authors are highly encouraged to include these. Given the discussions, I will keep my original score and lean towards acceptance.

---

> ### Author Response · Authors · 2021-08-31
> **Thanks to Reviewer tvph**
>
> We would like to thank the reviewer for taking the time to review our paper and for the valuable comments.
>
> Kindly let us know whether we have adequately addressed your comments on verifying our finding on feature-collision based attacks, the unclarity of our experimental setting, and the impact of the turning epoch on the weaker SIG attack. We truly appreciate your valuable feedback which helps verify the generality of our finding and improve the clarity of our experiments.

---

### Author Response · Authors · 2021-09-02
**Thanks to all reviewers**

We wish to thank you all for spending the time reviewing our work and your valuable feedback. Furthermore, we truly appreciate your patience in going through our responses and the follow-up discussion. We will carefully address your comments in the revision.

---

### Decision · Program_Chairs · 2021-09-27

**Decision:**

Accept (Poster)

**Comment:**

I read the paper and considered the discussions carefully, especially that of the authors with 9SWb who brought up many thoughtful comments and criticisms. I believe that the concerns and criticisms have been largely addressed by the authors during the rebuttal, which I will discuss below.

I will now discuss the merits of the paper:
1. Well-established threat model. The threat scenario of classical (non-federated) training on a dataset poisoned such that inserting the backdoor during inference will result in the target class is a well-established scenario. It is also one of the simplest scenarios, which means the attack is more likely to be deployed in the real world.
2. Novelty. To mine and all the reviewers' agreement, the observations and methods presented in this paper are quite novel. Compared to other backdoor defense methods which often aim to learn the backdoor pattern from among a large search space, I believe the easy-learning detection and backdoor unlearning methods are quite elegant since they exploit what seems to be a fundamental weakness of backdoors (that to be strong they must be easy to learn) and can be incorporated directly into the training procedure (as opposed to a separate stage).
3. Results. The authors have done a comprehensive study on 3 datasets and 6 backdoor attacks, and included some well-established datasets/architectures for comparison with prior art (i.e. resnet18 on cifar, along with resnet34 upon the request of 9SWb). The authors covered 4 additional attacks upon the request of reviewer 9SWb who subsequently raised their score from 3 to 5.
4. The authors provided ample ablation studies to provide better insight into why their method works, which helps the readers obtain some learnings from this paper.

There were some concerns about different threat models (e.g. adversary having control over more things than just a fraction of the dataset) and different training settings (federated learning). While interesting, these are outside the scope of the paper and that the current scope is sufficiently interesting.

I therefore recommend acceptance.